# PLSemanticsBench: Large Language Models as Programming Language Interpreters

## Abstract

As large language models (LLMs) excel at code reasoning, a natural question arises: can an LLM execute programs (i.e., act as an interpreter) purely based on a programming language's formal semantics? If so, it will enable rapid prototyping of new programming languages and language features. We study this question using the imperative language IMP (a subset of C), formalized via small-step operational semantics (SOS) and rewriting-based operational semantics (K-semantics). We introduce three evaluation sets—Human-Written, LLM-Translated, and Fuzzer-Generated-whose difficulty is controlled by code-complexity metrics spanning the size, control-flow, and data-flow axes. Given a program and its semantics formalized with SOS/K-semantics, models are evaluated on three tasks ranging from coarse to fine: (1) final-state prediction, (2) semantic rule prediction, and (3) execution trace prediction. To distinguish pretraining memorization from semantic competence, we define two nonstandard semantics obtained through systematic mutations of the standard rules. Across strong code/reasoning LLMs, performance drops under nonstandard semantics despite high performance under the standard one. We further find that (i) there are patterns to different model failures, (ii) most reasoning models perform exceptionally well on coarse grained tasks involving reasoning about highly complex programs often containing nested loop depths beyond five, and surprisingly, (iii) providing formal semantics helps on simple programs but often hurts on more complex ones. Overall, the results show a promise that LLMs could serve as programming language interpreters, but points to the lack of their robust semantics understanding.

## 1 Introduction

Programming language (PL) semantics formally defines the computational meaning of the program–i.e., how the program executes (Schmidt, 1996). It is common that the process of executing a program relies on an *interpreter*—a handcrafted engine that maps syntactic elements of a programming language to operational behavior defined by the PL semantics. Basically, the interpreter executes the given program step by step following the defined PL semantics rules. For decades, interpreters have served as indispensable tools in both the design and implementation of programming languages (Reynolds, 1972), enabling everything from debugging environments and educational tools to production systems. Yet despite their ubiquity, and unlike lexers and parsers (Appel, 1997), writing interpreters remains a labor-intensive (Peyton Jones, 1987; Aho & Johnson, 1976; Alfred et al., 2007), error-prone (Zang et al., 2024; Godefroid et al., 2008) task that requires deep expertise in programming languages and low-level execution models. This cost of development poses a challenge to the ongoing push to develop new domain-specific languages (Rocha Silva, 2022; Mernik et al., 2005) and enhance existing ones with new features (Castagna & Peyrot, 2025; Thimmaiah et al., 2024).

Large language models (LLMs) have shown promising performance in both code understanding and generation tasks such as code generation and code completion (Chen et al., 2021; El-Kishky et al., 2025; Team et al., 2023; Roziere et al., 2023; Zhang et al., 2022; Zhu et al., 2024; Hui et al., 2024). We ask the following question: whether LLMs truly understand the PL semantics and whether they are good enough to replace the handcrafted interpreters—i.e., to *simulate the operational behavior* of a program solely based on the PL semantics. If so, LLMs could be used (a) in early stages of rapid prototyping new programming languages or language features, (b) during debugging to understand

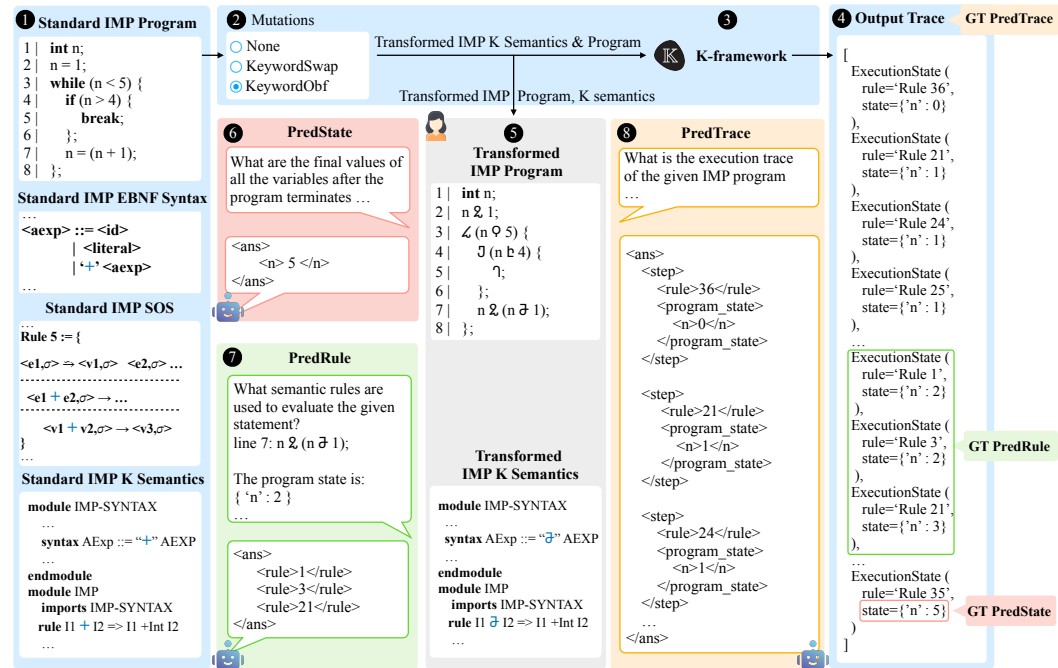

**Figure 1:** The PLSEMANTICSBENCH construction workflow and the proposed three tasks. Each program is written in IMP with syntax specified in EBNF, and its standard PL semantics defined using both SOS and K-semantics (❶). The standard semantics can be systematically transformed into one of two nonstandard semantics, KeywordSwap and KeywordObf (❷). The standard IMP programs and their semantics will be transformed accordingly. The transformed K-semantics is then used to build a traditional interpreter with the K-framework (❸), which generates an output trace (❹) for each transformed IMP program, serving as ground truth (GT) for the tasks. The transformed IMP program, its K-semantics (❺) are used to construct prompts for the tasks. Tasks (❻ - ❽) span from coarse-grained evaluation (PredState) to fine-grained evaluation (PredRule, PredTrace). An almost identical flow can also be achieved using the SOS and EBNF syntax by just replacing the K-framework interpreter with our custom built ANTLR4-based interpreter to evaluate the models on the tasks using SOS instead of K-semantics.

execution traces and program states, and (c) as a reference "implementation" for differential testing during development of the actual interpreter.

We introduce PLSEMANTICSBENCH, a benchmark designed to evaluate how LLMs handle code across distinct distributions. It includes a Human-Written split, reflecting natural programmer style, an LLM-Translated split, representing model-generated code, and a novel Fuzzer-Generated split. The fuzzer systematically produces rare control-flow patterns and edge-case semantics that are unlikely to appear in human code. Together, these datasets enable controlled and comprehensive evaluation of models on both realistic and adversarially challenging programs. Each split contains a number of programs written in the IMP language—a subset of C and a canonical imperative language used extensively in PL research—with the accompanying PL semantic rules. PLSEMANTICSBENCH focuses on probing an LLM's capability in serving as an interpreter which executes programs according to the specified PL semantics. As shown in Figure 1 (❶), each example in PLSEMANTICSBENCH consists of a program written in IMP, its syntax and the corresponding PL semantics specified formally using the small-step structural operational semantics (SOS) or rewriting-based operational semantics (K-semantics). Both SOS and K-semantics are included to evaluate robustness across different semantics styles.

**Task design**. PLSEMANTICSBENCH defines three tasks: *i) final-state prediction (PredState)*: predict the final program state (values of all the declared variables) under the given PL semantics (❻), *ii) semantic-rule prediction (PredRule)*: identify the ordered sequence of semantic rules required to evaluate the given statement (❼), *iii) execution-trace prediction (PredTrace)*: generate a step-by-step program execution trace, tuples of semantic rules and program states (❽). Each task targets a distinct aspect of the interpreter, collectively covering a broad spectrum of interpreter functionalities—from coarse-grained semantic check (PredState) to fine-grained symbolic execution (PredTrace).

```
Rule assgn_rred :=                  Rule assgn :=                      Rule decl :=
      ⟨e,σ⟩ → ⟨e',σ⟩
  ─────────────────────         ─────────────────────────        ────────────────────────────
  ⟨x = e;,σ⟩ → ⟨x = e';,σ⟩       ⟨x = v;,σ⟩ → ⟨ϵ,σ[x ↦ v]⟩         ⟨int x;,σ⟩ → ⟨ϵ,σ[x ↦ 0]⟩
Rule add_lred :=                    Rule add_rred :=                   Rule addition :=
      ⟨e1,σ⟩ → ⟨e1',σ⟩                   ⟨e2,σ⟩ → ⟨e2',σ⟩                 v3 = v1 + v2
  ─────────────────────────      ──────────────────────────       ──────────────────────
  ⟨e1 + e2,σ⟩ → ⟨e1' + e2,σ⟩      ⟨v1 + e2,σ⟩ → ⟨v1 + e2',σ⟩        ⟨v1 + v2,σ⟩ → v3
```

(a) Small-step (SOS) inference rules.

```
1  module SEMANTICS
2    imports SYNTAX //syntax is defined in a separate module, and looks similar to (a)
3    configuration <T> <k> $PGM:program </k> <state> .Map </state> </T>
4    rule <k> X = I:Int; => . ...</k> <state>... X |-> (_ => I) ...</state>
5    rule <k> int (X,Xs); ... </k> <state> Rho:Map (.Map => X|->0) </state>
6    rule <k> int .Ids; => . ...</k>
7    rule <k> X:Id => I ...</k> <state>... X |-> I ...</state>
8    rule I1 + I2 => I1 +Int I2
9  endmodule
```

(b) Rewriting rules as used in the K-framework.

**Figure 2:** The Small-step operational semantics (SOS) and rewriting-based operational semantics (K-semantics) for formalizing the semantics of a subset of the IMP programming language.

**Semantics mutation**. A critical challenge is to determine whether LLMs are truly interpreting programs based on the provided PL semantics, or merely relying on knowledge implicitly acquired during their pretraining (on popular programming languages). Specifically, the ability to generate functionally-correct programs or predict the outcomes of programs in the existing programming languages does not indicate an understanding of PL semantics, let alone acting as an interpreter. To address this challenge, we introduce two novel *semantic mutations* (❷) to derive the previously *unseen* nonstandard PL semantics from the standard one: 1) KeywordSwap ($s'_{ks}$): the semantic meanings of the operators are swapped (e.g., swap the semantic meanings of + and -), and 2) KeywordObf ($s'_{ko}$): common keywords and operators are replaced with rarely-seen symbols (e.g., using ⅎ instead of +). Success on tasks under nonstandard PL semantics requires a deep understanding of the PL semantics rather than just surface-level pattern matching.

We evaluate 11 state-of-the-art LLMs on PLSEMANTICSBENCH, covering models of various sizes, including both open-weight and closed-source models, as well as reasoning and non-reasoning models. Our findings show that LLMs generally perform well under standard PL semantics. Given the previously unseen nonstandard PL semantics, all models experience a decline across all the tasks compared to the standard one. The degradation is more noticeable in smaller and non-reasoning models. Reasoning models perform exceedingly well under standard semantics on the coarse grained task PredState with some of them passing the task on exceptionally complex programs involving nested loops with a nesting depth of five and greater. However, all models suffer on the fine-grained tasks PredRule and PredTrace. Overall, PLSEMANTICSBENCH reveals that models with strong performance on existing code generation benchmarks, such as BigCodeBench (Zhuo et al., 2024) and LiveCodeBench (Jain et al., 2024), does not imply that they possess an inherent understanding of PL semantics.

PLSEMANTICSBENCH is the first benchmark that evaluates the usability of LLMs as interpreters, laying the foundation for this novel line of research. Our empirical studies show that most state-of-the-art LLMs have a shallow understanding of PL semantics. We will publicly release the benchmark and supporting code after the review process.

## 2 BACKGROUND

**The IMP programming language.** IMP (a subset of C) is an imperative language that has been used extensively in PL research (Lesbre & Lemerre, 2024; Liu et al., 2024b). It supports the `int` and `bool` types, conditional statements (`if-else`), and looping constructs (`while`). It excludes functions and arrays for simplicity. We focus (in this section only) on a subset of IMP (only integer type, only literal addition expressions, variables can be re-declared, and no undeclared variables) to illustrate key concepts behind formalizing the semantics of a programming language.

**The semantics of a programming language** formally defines the behavior of its programs. In this work, we employ two styles for writing semantics.

***Structural operational semantics*** defines a language's behavior through inference rules that describe transitions between machine or program states. Key concepts include: 1) *Configurations*, representing the program and its execution context (e.g., the heap); 2) *Transitions*, denoting state changes driven by rule applications; and 3) *Inference rules*, specifying the semantics of language constructs. Depending on the granularity of transitions, operational semantics is categorized as small-step (SOS) or big-step. In SOS (Plotkin, 2004), each rule captures a minimal atomic computation step.

We now formalize the semantics of (a subset of) IMP using SOS. Table 1 gives a primer of the notations and their definitions which we use in our formalization. We use the configuration $\langle operation, \sigma \rangle$. The *operation* can be a statement or an expression. The $\sigma$ is the program state and maps a variable to an integer value. A one-step transition $\langle e, \sigma \rangle \rightarrow \langle e', \sigma \rangle$ implies that an expression e reduces to another expression e' through a single atomic computation step (e.g., `(1+1)+1` reducing to `2+1`).

The core SOS rules are shown in Figure 2a, using Gentzen-style inference notation (Gentzen, 1964). Each rule consists of premises, side conditions, and a conclusion: premises and side conditions appear above the fractional-line, and the conclusion below it. For example, the `assgn_rred` and `assgn` rules handle the assignment statement. The former has a premise that matches a compound integer addition expression which is reduced in its conclusion. This rule is applied repeatedly until the expression reduces to an integer literal which is then assigned to the variable by the latter. The rules for the addition operation

**Table 1:** Notation primer.

| Notation | Definition |
|---|---|
| $\sigma$ | Program state |
| x | Int variable |
| e | Int expression |
| v | Int literal |
| $\langle operation, \sigma \rangle$ | Configuration |
| $\sigma[x \mapsto v]$ | Store v in x |
| $\langle e, \sigma \rangle \rightarrow \langle e', \sigma \rangle$ | Transition |
| $\langle \epsilon, \sigma \rangle$ | NOP |

(`add_lred` and `add_rred`) similarly, reduce both the left and the right hand expressions until they reduce to integer literals. The `addition` rule is then applied to perform the addition operation. Our complete formalization of IMP using SOS is provided in Appendix A.

***Rewriting-based operational semantics*** (Roșu & Șerbănuță, 2010) is used in the K-framework, an executable semantic framework based on rewriting logic (Meseguer, 1992). K-framework is used for building interpreters given the syntax and semantics of programming languages. Figure 2b shows the semantics of the subset of IMP language defined using the K-semantics. The `SEMANTICS` module imports the `SYNTAX` module (line 2, omitted for brevity). The configuration (line 3) models the program state as a map-based store. Semantics is defined via rewrite rules (lines 4-8) that apply when their precondition patterns match.

## 3 BENCHMARK CONSTRUCTION

An overview of the benchmark construction process is shown in Figure 1. We formalize IMP in both SOS and K-semantics (❶). On experiments with K-semantics, we use the K-framework (❸) to obtain the ground-truths (❹) for all the tasks (we use our custom built ANTLR4-based interpreter for SOS). The IMP program along with the K-semantics (or SOS) is used to prompt the LLMs (❺). The rest of the section details the curation of the three datasets (Section 3.1) and the derivation of nonstandard semantics (Section 3.2).

### 3.1 DATASET CURATION

PLSEMANTICSBENCH contains three datasets namely, the Human-Written, the LLM-Translated, and the Fuzzer-Generated.

**Human-Written**. The IMP programs are manually adapted from C++ solutions to coding problems sourced from LeetCode (LeetCode, 2024), HumanEval (Chen et al., 2021; Zheng et al., 2023), CodeContests (Li et al., 2022), and MBPP (Austin et al., 2021; Orlanski et al., 2023). We use public test cases as input and their corresponding oracles as expected outcomes. C++ programs with `for` loops are rewritten to `while` loops to match IMP's capabilities. Additionally, we obfuscate variable names by replacing semantically meaningful identifiers (e.g., `maxIter`) with random strings (e.g., `a`). We show one example `C++` and IMP program in Appendix B.1. To validate correctness, we execute the IMP programs using K-framework and verify that the outputs match the test oracles.

**LLM-Translated**. The IMP programs are translated from C++ programs using LLMs. Specifically, we collect the C++ programs from the CodeForces solutions published on Hugging Face (Penedo et al., 2025). We prompt QWEN2.5-INSTRUCT 32B with the IMP syntax, semantics constraints, the

**Table 2:** Median code-complexity statistics summarizing the datasets used in our experiments. Control-flow complexity is characterized using extended cyclomatic complexity ($\Omega_{CC}$), maximum nested if–else ($\Omega_{If}$) and nested loop ($\Omega_{Loop}$) depths , maximum taken nested if–else ($\hat{\Omega}_{If}$), and taken nested loop ($\hat{\Omega}_{Loop}$) depths . Data-flow complexity is analyzed using DepDegree($\Omega_{DD}$) and the total number of assignments to variables in execution traces ($\hat{\Omega}_{Assign}$). Program size complexity is measured using lines of code ($\Omega_{Loc}$), Halstead metrics Volume ($\Omega_{Vol}$) and Vocabulary($\Omega_{Voc}$), and execution trace length ($\hat{\Omega}_{Trace}$). All metrics computed under dynamic-analysis are shown with a hat.

| Dataset | #Prog | Control-flow | | | | | Data-flow | | Size | | | |
|---|---|---|---|---|---|---|---|---|---|---|---|---|
| | | $\Omega_{CC}$ | $\Omega_{If}$ | $\Omega_{Loop}$ | $\hat{\Omega}_{If}$ | $\hat{\Omega}_{Loop}$ | $\Omega_{DD}$ | $\hat{\Omega}_{Assign}$ | $\Omega_{Loc}$ | $\Omega_{Vol}$ | $\Omega_{Voc}$ | $\hat{\Omega}_{Trace}$ |
| Human-Written | 162 | 3 | 1 | 1 | 1 | 1 | 12 | 9 | 19 | 320 | 22 | 20 |
| LLM-Translated | 165 | 9 | 1 | 1 | 1 | 1 | 48 | 62 | 106 | 2K | 35 | 180 |
| Fuzzer-Generated | 165 | 100 | 7 | 6 | 2 | 1 | 6K | 86 | 794 | 63K | 112 | 190 |

C++ solution and one corresponding public test case to instruct it to generate a valid IMP program. We filter the generated IMP programs with the K-framework to retain only those that are executable and have normal termination.

**Fuzzer-Generated**. We construct this with a depth-controlled, semantics-aware, grammar-based fuzzer (Yang et al., 2011; Han et al., 2019); a fuzzer is a tool that automatically generates programs and it is commonly used for testing compilers and interpreters. At each block, the fuzzer samples a statement from {`assign`, `if-else`, `while`, `break`, `continue`, `halt`} using depth-tapered probabilities—a cosine decay reduces the chance of generating new `if`/`while` as nesting grows—and legality masks that forbid `break`/`continue` outside loops. To encourage termination, every `while` is instrumented with a private loop-breaker variable that is monotonically updated in the body and whose bound is conjoined with the loop predicate (`cond ∧ bound`). More details about the fuzzer's settings and the generated IMP programs is discussed in Appendix B.2.

**Program complexity and data statistics**. We characterize program complexity along three axes—control-flow, data-flow, and size. For control-flow, we use extended cyclomatic complexity ($\Omega_{CC}$) (McCabe, 1976); the *static* maximum nesting depths of if–else and while ($\Omega_{If}$, $\Omega_{Loop}$); and their *dynamic* counterparts measured along executed paths ($\hat{\Omega}_{If}$, $\hat{\Omega}_{Loop}$). For data-flow, we use DepDegree ($\Omega_{DD}$), which quantifies uses and redefinitions of declared variables (Beyer & Fararooy, 2010), and the total number of executed assignments ($\hat{\Omega}_{Assign}$). For size, we use Halstead Vocabulary and Volume ($\Omega_{Voc}$, $\Omega_{Vol}$) (Halstead, 1977) which captures the symbol variety and program information in bits respectively, lines of code ($\Omega_{Loc}$), and execution-trace length ($\hat{\Omega}_{Trace}$) under SOS.

Table 2 reports median values of the complexity metrics per dataset. Across ∼165 programs per split, the median complexity increases progressively from Human-Written to LLM-Translated to Fuzzer-Generated along all three axes. The distributions of these complexity metrics for the three datasets is given in Appendix C.

## 3.2 Nonstandard Semantics

We introduce two nonstandard semantics, KeywordSwap and KeywordObf, to assess the models' ability to truly interpret the programs according to the provided PL semantics rather than relying on the knowledge obtained during training on large existing code corpora. These nonstandard semantics are derived from the standard IMP PL semantics through operator and keyword mutations and obfuscations.

**KeywordSwap** ($s'_{ks}$). We derive KeywordSwap by swapping the semantic meanings of the syntactic operators in the standard semantics to their KeywordSwap counterparts as shown in Table 3. For example, KeywordSwap swaps the semantics of the addition (+) and the subtraction (-) operators. Therefore, an integer addition expression (e.g., `x+y`) under the standard IMP semantics is evaluated as if it were a subtraction expression (e.g., `x-y`) under KeywordSwap.

**KeywordObf** ($s'_{ko}$). We derive KeywordObf through obfuscation by replacing keywords and operators in the standard semantics with characters from the rare Caucasian-Albanian script (Gippert & Schulze, 2023). Some of the obfuscations used are shown in Table 3, which replaces the syntactic operators and keywords defined in the standard semantics with their KeywordObf counterparts. After applying this obfuscation, the expression (e.g., `x ꕕ y`) under KeywordObf would execute identically as the integer addition expression (e.g., `x+y`) under the standard IMP semantics.

**Table 3:** Some of the mutations and obfuscations applied to the standard semantics to derive the nonstandard semantics KeywordSwap and KeywordObf. The complete list is given in Appendix D.4.

| Type | Arithmetic | | | | | Relational | | | | | | Logical | | | Keyword |
|------|---|---|---|---|---|---|---|---|---|---|---|---|---|---|---|
| Standard | + | − | ⋆ | / | % | < | <= | > | >= | == | != | ! | && | \|\| | while |
| KeywordSwap | − | + | / | ⋆ | % | > | >= | < | <= | != | == | ! | \|\| | && | while |
| KeywordObf | ꝺ | ꞁꞁ | ꝗ | ꝰ | ꝳ | ꝯ | ꝑ | �040 | ꞃ | ꝥ | ꝙ | ꝩ | ꝅ | ꞈ | ꝇ |

The KeywordSwap nonstandard semantics explores the impact of pretraining bias (e.g., redefining, the typically encountered mapping of the symbol (+) to addition operation) on LLMs' understanding of PL semantics by swapping the semantic meanings of standard operators, while retaining the familiar symbols. In contrast, the KeywordObf nonstandard semantics examines LLMs' performance in the context of mitigating pretraining bias. This is achieved by obfuscating standard operators and keywords with symbols from the rarely encountered Caucasian-Albanian script.

# 4 EXPERIMENTS

**Table 4:** Evaluated Models.

| Model | Reference |
|-------|-----------|
| LLAMA-3.3 70B | Grattafiori et al. (2024) |
| QWEN2.5-INSTRUCT 14B | Hui et al. (2024) |
| QWEN2.5-INSTRUCT 32B | |
| GPT-4O-MINI | Achiam et al. (2023) |
| O3-MINI | OpenAI (2025) |
| GPT-5-MINI | |
| DEEPSEEK-LLAMA 70B | |
| DEEPSEEK-QWEN 14B | Guo et al. (2025) |
| DEEPSEEK-QWEN 32B | |
| QWQ 32B | Team (2025b) |
| GEMINI-2.5-PRO | Kavukcuoglu (2025) |

**Evaluation settings**. We benchmark each model under six semantics–program configurations: $(s, p)$ (standard semantics and program), $(s'_{ks}, p'_{ks})$ (KeywordSwap), and $(s'_{ko}, p'_{ko})$ (KeywordObf), each instantiated for both SOS and the K-semantics variants. For the PredState task we additionally report a *no-semantics baseline* $(p)$ that provides only the standard program. Table 4 shows the code-centric non-reasoning and reasoning models used in our experiments. For non-reasoning models we report both direct (no-CoT) and CoT prompting (explain step-by-step, then answer).

**Datasets and tasks**. We evaluate all models on the Human-Written split for all tasks. For the more complex LLM-Translated and Fuzzer-Generated splits, we restrict evaluation to the best-performing models on the PredState task (top-3 from different families by Human-Written accuracy) for several reasons: (i) PredState task performance on Human-Written is near-saturated, motivating evaluation on harder distributions; (ii) PredRule task is largely agnostic to program complexity (see Appendix E.2.1); (iii) performance on PredTrace task remains uniformly low even on Human-Written split, offering limited additional insight on harder splits; and (iv) reduce cost of experiments. We only average all reasoning model experiments (and GPT-4O-MINI) over three runs. The temperature for all the non-reasoning models (except GPT-4O-MINI) is set to 0. Prompt templates and experiment details are given in Appendix D.

## 4.1 FINAL-STATE PREDICTION (PREDSTATE)

**Task**. As a coarse-grained measure of LLMs' performance as an interpreter, we challenge them with predicting the final states of all the declared variables in a given program (Figure 1 ❻). We explore this under the cases when no-semantics is provided and when the semantics are provided using the K-semantics and SOS styles.

**Data curation and results**. The IMP programs in all the three datasets are executed with the K-framework to obtain the gold execution traces. Every element in the execution trace is a tuple of a semantic rule (K-semantics or SOS) needed to evaluate a statement and the program state (values of all declared variables) after executing that rule. Thus the state of the final element from the execution trace is used as the ground-truth for the PredState task. Table 5 shows the accuracies of the models on the PredState task. More details, such as the average percentage of variables predicted correctly etc., is discussed in Appendix E.1.3.

*Does providing semantics help?* On the Human-Written dataset we see that providing semantics (K-semantics or SOS) generally hurts the performance of non-reasoning models but significantly improves the performance of reasoning models. The trend is similar on the LLM-Translated dataset but to a lesser extent, while in the Fuzzer-Generated dataset, the trend reverses and providing semantics hurts the performances of even the reasoning models.

**Table 5:** Accuracies of the models on the PredState task, using SOS and K-semantics, for both the standard and nonstandard variants across the Human-Written, the LLM-Translated, and the Fuzzer-Generated datasets. The cases where models under standard semantics perform better/worse than with no-semantics are shaded green/red.

| | Models | $p$ | K-semantics | | | SOS | | |
|---|---|---|---|---|---|---|---|---|
| | | | $(s,p)$ | $(s'_{ks}, p'_{ks})$ | $(s'_{ko}, p'_{ko})$ | $(s,p)$ | $(s'_{ks}, p'_{ks})$ | $(s'_{ko}, p'_{ko})$ |
| | **Human-Written** | | | | | | | |
| Non-reasoning | QWEN2.5-INSTRUCT 14B | 33 | 27 | 6 | 14 | 28 | 6 | 8 |
| | QWEN2.5-INSTRUCT 14B-CoT | 73 | 70 | 2 | 48 | 68 | 4 | 41 |
| | QWEN2.5-INSTRUCT 32B | 50 | 29 | 4 | 12 | 33 | 4 | 19 |
| | QWEN2.5-INSTRUCT 32B-CoT | 81 | 77 | 8 | 56 | 69 | 3 | 33 |
| | LLAMA-3.3 70B | 32 | 29 | 4 | 12 | 25 | 5 | 12 |
| | LLAMA-3.3 70B-CoT | 75 | 75 | 3 | 56 | 77 | 2 | 48 |
| | GPT-4O-MINI | 31 | 26 | 6 | 8 | 24 | 6 | 8 |
| | GPT-4O-MINI-CoT | 68 | 78 | 2 | 38 | 65 | 3 | 27 |
| Reasoning | DEEPSEEK-QWEN 14B | 65 | 81 | 2 | 40 | 58 | 2 | 29 |
| | DEEPSEEK-QWEN 32B | 84 | 93 | 21 | 72 | 95 | 3 | 77 |
| | DEEPSEEK-LLAMA 70B | 80 | 88 | 2 | 58 | 89 | 2 | 59 |
| | QWQ 32B | 93 | 98 | 71 | 82 | 98 | 7 | 86 |
| | O3-MINI | 94 | **100** | 41 | 84 | **100** | 63 | 95 |
| | GPT-5-MINI | **100** | 99 | 79 | 94 | 100 | 79 | 99 |
| | GEMINI-2.5-PRO | 93 | 100 | **97** | **94** | 99 | **98** | **100** |
| | **LLM-Translated** | | | | | | | |
| | QWQ 32B | 82 | 83 | 31 | 61 | 82 | 4 | 63 |
| | GPT-5-MINI | **94** | **96** | 76 | 86 | **95** | 65 | 90 |
| | GEMINI-2.5-PRO | 91 | 94 | **85** | **91** | 94 | **87** | **93** |
| | **Fuzzer-Generated** | | | | | | | |
| | QWQ 32B | 16 | 16 | 0 | 3 | 15 | 0 | 1 |
| | GPT-5-MINI | 57 | 51 | 14 | 23 | 55 | 17 | 23 |
| | GEMINI-2.5-PRO | **73** | **69** | 26 | 49 | **69** | 39 | 47 |

*How well do models perform on nonstandard semantics?* On all the datasets, models perform better under standard than under nonstandard semantics (only exception is GEMINI-2.5-PRO under SOS on the Human-Written split). For the nonstandard semantics, the models perform significantly better with KeywordObf than KeywordSwap. Only GEMINI-2.5-PRO performs on par for both the nonstandard semantics'. Manual inspection of the KeywordSwap failure samples indicated models failing to use the re-defined semantics of the well known operators (e.g., re-defining (+) as subtraction) as the primary reason for poor performance.

*Which code-complexity metrics best predict LLM mispredictions?* To answer this, we train an Elastic Net logistic regression classifier using the IMP programs' complexity metrics as features and the LLMs' pass/fail outcomes as labels. The regression coefficients are transformed into odds ratios per interquartile range, $\Theta(\Delta)$, which quantify how the odds of success change as a metric increases from its $25^{th}$ to $75^{th}$ percentile, holding all other metrics constant. A negative $\Theta(\Delta)$ indicates performance degradation, while a positive value suggests improvement. Our analysis (Appendix E.1.1) shows that nearly all metrics consistently correlate with worse performance as they increase. In particular, deeper control-flow structures most strongly harm accuracy on human-written code, whereas larger data-flow and size-related metrics dominate the degradation on code translated and generated by LLMs and fuzzers repectively.

*Is there a systematic pattern in how complexity metrics impact different models?* To investigate, we apply hierarchical clustering (Johnson, 1967) to the standardized regression coefficients (from the logistic regression analysis) across metrics. We then use a one-vs-rest Cohen's d test (Cohen, 1988) to identify the two most distinguishing metrics for each cluster. This analysis (Appendix E.1.2) reveals three clear groups: (i) non-reasoning models without CoT prompting, (ii) primarily reasoning and CoT-augmented non-reasoning models under K-semantics, and (iii) reasoning and CoT-augmented non-reasoning models under SOS semantics.

## 4.2 SEMANTIC-RULE PREDICTION (PREDRULE)

**Task**. Traditional interpreters follow predefined semantic rules to execute programs. The PredRule task evaluates whether LLMs can correctly select the specific PL semantic rules to execute the program. Given the PL semantics, a program statement, and the program state (variables and their

**Table 6:** The exact-match accuracies of the models on the semantic-rule prediction task under SOS and K-semantics on the Human-Written dataset. The cases where the models under one of the nonstandard semantics perform better/worse relative to their counterpart are shaded green/red.

| | Models | K-semantics | | | SOS | | |
|---|---|---|---|---|---|---|---|
| | | $(s,p)$ | $(s'_{ks}, p'_{ks})$ | $(s'_{ko}, p'_{ko})$ | $(s,p)$ | $(s'_{ks}, p'_{ks})$ | $(s'_{ko}, p'_{ko})$ |
| Non-reasoning | LLAMA-3.3 70B | 45 | 42 | 45 | 32 | 32 | 27 |
| | LLAMA-3.3 70B-CoT | 69 | 46 | 50 | 28 | 28 | 17 |
| | QWEN2.5-INSTRUCT 14B | 49 | 45 | 45 | 19 | 19 | 17 |
| | QWEN2.5-INSTRUCT 14B-CoT | 50 | 32 | 27 | 12 | 10 | 6 |
| | QWEN2.5-INSTRUCT 32B | 58 | 52 | 46 | 17 | 24 | 19 |
| | QWEN2.5-INSTRUCT 32B-CoT | 64 | 47 | 47 | 29 | 26 | 24 |
| | GPT-4O-MINI | 38 | 34 | 27 | 27 | 27 | 21 |
| | GPT-4O-MINI-CoT | 57 | 46 | 37 | 27 | 26 | 24 |
| Reasoning | DEEPSEEK-QWEN 14B | 57 | 45 | 48 | 22 | 21 | 20 |
| | DEEPSEEK-QWEN 32B | 79 | 66 | 65 | 47 | 38 | 38 |
| | DEEPSEEK-LLAMA 70B | 34 | 10 | 27 | 1 | 1 | 1 |
| | GEMINI-2.5-PRO | **99** | **98** | **90** | **94** | **96** | **98** |
| | O3-MINI | 93 | 65 | 84 | 80 | 72 | 67 |
| | QwQ 32B | 92 | 85 | 76 | 49 | 44 | 41 |
| | GPT-5-MINI | 92 | 83 | 82 | 80 | 81 | 81 |

values) before the statement's execution, the LLMs are expected to predict the correct sequence of semantic rules, both in terms of the rules and their application order, to accurately evaluate the statement. We show one example of a model's expected output in Figure 1 (❼). Some statements may require just a single rule whereas others may need a sequence of several rules.

**Data curation and results**. To obtain the ground-truth list of K-semantics and SOS rules, we execute the IMP programs with the K-framework and our ANTLR4-based interpreter respectively. For each program, we select a subset of statements for evaluation. To balance diversity and the number of chosen statements, we group together statements requiring identical sequences of semantic rules and randomly select one from each group, with a maximum of 10 statements per program. Table 6 shows the exact-match accuracy, i.e., the percentage of predicted semantic rule sequences that exactly match the ground-truth sequences.

*How does the models' performances compare between the K-semantics and SOS?* From Table 6 we see that models perform slightly better when provided with K-semantics relative to SOS. This could be due to two contributing factors: 1) SOS on average requires more rules (e.g., left reduction, right reduction and the application of the operator itself are all different rules for the addition operation in SOS, whereas it is just a single rule in K-semantics) to evaluate a statement than its K-semantics counterpart (see Appendix E.2.2), and 2) several large examples of formalizing languages such as C (K Framework Team, 2025a), Java (K Framework Team, 2025b), and Python (Runtime Verification, 2025) etc. exist for K-semantics but none for SOS, therefore models may be more familiar with K-semantics than SOS.

*How does the models' performances compare for the nonstandard semantics?* For the non-reasoning models, the performances are consistently better for KeywordSwap under SOS and generally better for KeywordSwap under K-semantics than their corresponding KeywordObf counterparts. On the other hand, the differences in performances between the two nonstandard semantics for the reasoning models is less apparent under both, K-semantics and SOS. Furthermore, with the exception of the DEEPSEEK-LLAMA 70B model, reasoning models outperform the non-reasoning ones for all the cases.

## 4.3 EXECUTION-TRACE PREDICTION (PREDTRACE)

**Task**. In addition to executing individual statements, an interpreter maintains the program state and determines the next statement to execute throughout a program's execution. The PredTrace task challenges LLMs to predict the complete execution trace, which is defined as an ordered sequence of *execution steps*. Each step is a tuple of a semantic rule (K-semantics or SOS needed to evaluate the statement being executed currently) and the program state after applying the rule. An example of a predicted execution trace is given in Figure 1 (❽).

**Data curation and results**. We use the K-framework and our ANTLR4-based IMP interpreter to generate execution traces for the K-semantics and SOS variants respectively—which we post-process

**Table 7:** The exact-match accuracies of the models on the execution-trace prediction task under SOS and K-semantics on the Human-Written dataset. All non-reasoning models (not shown) scored near zero.

| | Models | K-semantics | | | SOS | | |
|---|---|---|---|---|---|---|---|
| | | $(s,p)$ | $(s'_{ks}, p'_{ks})$ | $(s'_{ko}, p'_{ko})$ | $(s,p)$ | $(s'_{ks}, p'_{ks})$ | $(s'_{ko}, p'_{ko})$ |
| Reasoning | DEEPSEEK-QWEN 14B | 1 | 0 | 0 | 0 | 0 | 0 |
| | DEEPSEEK-QWEN 32B | 8 | 2 | 3 | 0 | 0 | 1 |
| | DEEPSEEK-LLAMA 70B | 3 | 0 | 3 | 0 | 0 | 0 |
| | GEMINI-2.5-PRO | **25** | **25** | **25** | 32 | 35 | 35 |
| | O3-MINI | 19 | 3 | 13 | 5 | 3 | 2 |
| | QWQ 32B | 18 | 16 | 15 | 0 | 0 | 0 |
| | GPT-5-MINI | 20 | 14 | 17 | 17 | 15 | 17 |

into XML. Table 7 shows the exact-match accuracies across models. All the models perform poorly on the PredTrace task.

*How do non-reasoning models compare against reasoning models?* The peformance of non-reasoning models is significantly worse than their reasoning counterparts. Most of the non-reasoning models with the exception of LLAMA-3.3 70B-family of models, fail to correctly predict the complete execution trace for even a single program in the Human-Written dataset. Reasoning capability is therefore observed to be an important factor in understanding of the semantics and program interpretation.

*How do performances on K-semantics compare against SOS?* Both non-reasoning and reasoning models perform better on K-semantics than on SOS. Most models score near zero under SOS. This could be due to them being more familiar with K-semantics formalization structure and due to the execution trace lengths under SOS being longer. The only exception is the GEMINI-2.5-PRO model which consistently performs better under SOS semantics and is also the best performing model in this task.

## 5 RELATED WORK

**Code reasoning and execution benchmarks**. Several benchmarks assess LLMs' ability to reason about program execution. CRUXeval (Gu et al., 2024) evaluates test output prediction for Python programs. LiveCodeBench (Jain et al., 2024) adds test prediction and program repair. REval (Chen et al., 2024) tests understanding of runtime behavior via program states, paths, and outputs. Coconut (Beger & Dutta, 2025) targets control-flow reasoning by predicting execution line sequences, and CodeMind (Liu et al., 2024a) introduces inductive program-simulation tasks. Most recently, CWM (Team, 2025a) releases a 32B open-weights LLM for code generation with world-model style training on execution traces, aiming to internalize program dynamics. However, none of these benchmarks are designed to evaluate LLMs strictly as interpreters of user-defined PL semantics.

**Semantics-oriented training and evaluation**. SemCoder (Ding et al., 2024) trains LLMs on symbolic, operational, and abstract semantics tasks. SpecEval (Ma et al., 2024) evaluates semantic understanding of JML specifications, while LMS (Ma et al., 2023) tests structural recovery of ASTs and CFGs. Other efforts, such as CodeARC (Wei et al., 2025b) and EquiBench (Wei et al., 2025a), study robustness under semantic-preserving mutations. In contrast, our benchmark frames evaluation as an interpreter task, requiring models to execute programs according to formal semantics (SOS) and its variants.

To our knowledge, this is the first work to measure executability, trace simulation, and rule-level reasoning in a unified, semantics-driven framework.

## 6 CONCLUSION

We introduced PLSEMANTICSBENCH, the first benchmark for evaluating LLMs as PL interpreters guided by formal semantics. The benchmark spans three dataset splits, two semantic variants, and three tasks that probe different dimensions of interpreter functionality. While some LLMs achieve strong performance on coarse-grained tasks and simpler programs—and can even generalize across different semantic rule notations such as SOS —we uncover substantial gaps on fine-grained tasks, nonstandard semantics, and complex programs. These findings highlight both the promise and the current limitations of semantics-aware LLMs. Looking forward, we believe that explicitly teaching language semantics to LLMs can pave the way for rapid prototyping of new programming languages and the extension of existing ones.

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

**Appendix**

## A  IMP FORMALIZATION

Here we describe the syntax and the semantics rules for IMP used in all our experiments.

### A.1  IMP SYNTAX DESCRIPTION

The IMP syntax used in all our experiments is given in EBNF in Figure 3. The terminals are shown in red while the non-terminals are shown in blue.

```
1   <program>    ::= <stmt_list>
2   <stmt_list> ::= (<stmt> ';')*
3   <stmt>       ::= 'int' <id>
4                 | <id> '=' <aexp>
5                 | 'if' '(' <bexp> ')' '{' <stmt_list> '}' 'else' '{' <stmt_list> '}'
6                 | 'while' '(' <bexp> ')' '{' <stmt_list> '}'
7                 | 'loop' '(' <bexp> ')' '{' <stmt_list> '}'
8                 | 'halt'
9                 | 'continue'
10                | 'break'
11                | 'LE'
12  <aexp>       ::= <id>
13                | <literal>
14                | '(' <aexp>? <mathop> <aexp> ')'
15  <bexp>       ::= '(' <bool> ')'
16                | '(' <aexp> <relop> <aexp> ')'
17                | '(' <lognot> <bexp> ')'
18                | '(' <bexp> <logicalop> <bexp> ')'
19  <bool>       ::= 'true' | 'false'
20  <mathop>     ::= '+' | '-' | '*' | '/' | '%'
21  <relop>      ::= '<' | '<=' | '>' | '>=' | '==' | '!='
22  <lognot>     ::= '!'
23  <logicalop> ::= '&&' | '||'
24  <id>         ::= <letter>+
25  <literal>    ::= <digit>+
26  <letter>     ::= 'a' | 'b' | 'c' | 'd' | 'e' | 'f' | 'g' | 'h' | 'i' | 'j'
27                | 'k' | 'l' | 'm' | 'n' | 'o' | 'p' | 'q' | 'r' | 's' | 't'
28                | 'u' | 'v' | 'w' | 'x' | 'y' | 'z'
29                | 'A' | 'B' | 'C' | 'D' | 'E' | 'F' | 'G' | 'H' | 'I' | 'J'
30                | 'K' | 'L' | 'M' | 'N' | 'O' | 'P' | 'Q' | 'R' | 'S' | 'T'
31                | 'U' | 'V' | 'W' | 'X' | 'Y' | 'Z'
32  <digit>      ::= '0' | '1' | '2' | '3' | '4' | '5' | '6' | '7' | '8' | '9'
```

**Figure 3:** Complete syntax of IMP used in our experiments in EBNF.

### A.2  SMALL-STEP OPERATIONAL SEMANTICS RULES FOR IMP

**Table 8:** Metavariables used in the SOS formalization of IMP.

| Meta-var | Sort | Ranges over / Domain |
|---|---|---|
| x | id | Identifiers (program variable names) |
| v | literal | Integer literals |
| q | bool | Boolean literals |
| a | aexp | Integer expressions |
| b | bexp | Boolean expressions |
| s | stmt | Statements of the language |
| SL | stmt_list | Finite statement lists (SL ::= $\epsilon$ \| s :: SL') |

We formalize IMP using a small-step structural operational semantics (SOS). A configuration is a triple

$$\langle \text{operation}, \sigma, \chi \rangle,$$

where $\sigma : \text{id} \mapsto \text{literal}$ is the program store mapping identifiers to values, and $\chi$ is a last-in, first-out *control stack* of loop headers that records the dynamic nesting of currently active loops:

$$\chi ::= \epsilon \mid \text{s} :: \chi'.$$

The top of $\chi$ is the innermost executing loop.

We use standard metavariables $x, v, q, a, b, s, \text{SL}$ with their sorts summarized in Table 8. For example, a ranges over arithmetic expressions, so rules mentioning a1, a2, ... concern arithmetic

**Table 9:** Metafunctions for control stack and statement-list concatenation.

| Function | Signature | Definition |
|---|---|---|
| push | $\text{stmt} \times \text{Stack} \to \text{Stack}$ | $\text{push}(s, \chi) \triangleq s :: \chi$ |
| pop | $\text{Stack}_{\neq \epsilon} \to \text{Stack}$ | $\text{pop}(s :: \chi) \triangleq \chi$ |
| top | $\text{Stack} \to \text{stmt} \cup \{\epsilon\}$ | $\text{top}(\chi) \triangleq \begin{cases} \epsilon & \text{if } \chi = \epsilon, \\ s & \text{if } \chi = s :: \chi' \end{cases}$ |
| ++ | $\text{stmt\_list} \times \text{stmt\_list} \to \text{stmt\_list}$ | $SL1 ++ SL2 \triangleq \begin{cases} SL2 & \text{if } SL1 = \epsilon, \\ s :: (SL1' ++ SL2) & \text{if } SL1 = s :: SL1'. \end{cases}$ |

evaluation. Auxiliary metafunctions for manipulating the control stack (push, pop, top) and concatenating statement lists ( ++ ) are given in Table 9.

Program execution proceeds by repeatedly applying the transition relation $\to$ to configurations, starting from $\langle \text{SL}, \sigma, \chi \rangle$, where SL is the program's statement list, until a terminal configuration is reached. We treat $\langle \epsilon, \sigma, \chi \rangle$, $\langle \text{halt}, \sigma, \chi \rangle$, and $\langle \text{ERROR}, \sigma, \chi \rangle$ as terminal.

The complete set of small-step SOS rules defining the semantics of IMP appears in Table 10.

**Table 10:** Small-step SOS rules used to formalize IMP.

| Rule | Formalization | Description |
|---|---|---|
| Rule 1 | $$\frac{\sigma(\text{x}) = \text{v}}{\langle \text{x}, \sigma, \chi \rangle \to \text{v}}$$ | Variable lookup returns value. |
| Rule 2 | $$\frac{\sigma(\text{x}) = \bot}{\langle \text{x}, \sigma, \chi \rangle \to \langle \text{ERROR}, \sigma, \chi \rangle}$$ | Read of undefined variable errors. |
| Rule 3 | $$\overline{\langle \text{int x ::  SL}, \sigma, \chi \rangle \to \langle \text{SL}, \sigma[\text{x} \mapsto 0], \chi \rangle}$$ | Declared int variable initialized to 0. |
| Rule 4 | $$\frac{\langle \text{a}, \sigma, \chi \rangle \to \langle \text{a}', \sigma, \chi \rangle}{\langle \text{x := a ::  SL}, \sigma, \chi \rangle \to \langle \text{x := a' ::  SL}, \sigma, \chi \rangle}$$ | Assignment expression steps. |
| Rule 5 | $$\frac{\sigma(\text{x}) \neq \bot}{\langle \text{x := v ::  SL}, \sigma, \chi \rangle \to \langle \text{SL}, \sigma[\text{x} \mapsto \text{v}], \chi \rangle}$$ | Writeback to existing variable. |
| Rule 6 | $$\frac{\sigma(\text{x}) = \bot}{\langle \text{x := v ::  SL}, \sigma, \chi \rangle \to \langle \text{ERROR}, \sigma, \chi \rangle}$$ | Assign to undefined variable errors. |
| Rule 7 | $$\frac{\langle \text{a1}, \sigma, \chi \rangle \to \langle \text{a1}', \sigma, \chi \rangle}{\langle \text{a1 + a2}, \sigma, \chi \rangle \to \langle \text{a1' + a2}, \sigma, \chi \rangle}$$ | Plus — step left operand. |
| Rule 8 | $$\frac{\langle \text{a2}, \sigma, \chi \rangle \to \langle \text{a2}', \sigma, \chi \rangle}{\langle \text{v1 + a2}, \sigma, \chi \rangle \to \langle \text{v1 + a2'}, \sigma, \chi \rangle}$$ | Plus — step right operand. |
| Rule 9 | $$\frac{\text{v3} = \text{v1} + \text{v2}}{\langle \text{v1 + v2}, \sigma, \chi \rangle \to \text{v3}}$$ | Plus — compute. |

| Rule 10 | $$\frac{\langle \texttt{a1}, \sigma, \chi \rangle \rightarrow \langle \texttt{a1'}, \sigma, \chi \rangle}{\langle \texttt{a1 - a2}, \sigma, \chi \rangle \rightarrow \langle \texttt{a1' - a2}, \sigma, \chi \rangle}$$ | Minus — step left operand. |
|---|---|---|
| Rule 11 | $$\frac{\langle \texttt{a2}, \sigma, \chi \rangle \rightarrow \langle \texttt{a2'}, \sigma, \chi \rangle}{\langle \texttt{v1 - a2}, \sigma, \chi \rangle \rightarrow \langle \texttt{v1 - a2'}, \sigma, \chi \rangle}$$ | Minus — step right operand. |
| Rule 12 | $$\frac{\texttt{v3} = \texttt{v1} - \texttt{v2}}{\langle \texttt{v1 - v2}, \sigma, \chi \rangle \rightarrow \texttt{v3}}$$ | Minus — compute. |
| Rule 13 | $$\frac{\langle \texttt{a1}, \sigma, \chi \rangle \rightarrow \langle \texttt{a1'}, \sigma, \chi \rangle}{\langle \texttt{a1 * a2}, \sigma, \chi \rangle \rightarrow \langle \texttt{a1' * a2}, \sigma, \chi \rangle}$$ | Times — step left operand. |
| Rule 14 | $$\frac{\langle \texttt{a2}, \sigma, \chi \rangle \rightarrow \langle \texttt{a2'}, \sigma, \chi \rangle}{\langle \texttt{v1 * a2}, \sigma, \chi \rangle \rightarrow \langle \texttt{v1 * a2'}, \sigma, \chi \rangle}$$ | Times — step right operand. |
| Rule 15 | $$\frac{\texttt{v3} = \texttt{v1} * \texttt{v2}}{\langle \texttt{v1 * v2}, \sigma, \chi \rangle \rightarrow \texttt{v3}}$$ | Times — compute. |
| Rule 16 | $$\frac{\langle \texttt{a1}, \sigma, \chi \rangle \rightarrow \langle \texttt{a1'}, \sigma, \chi \rangle}{\langle \texttt{a1 / a2}, \sigma, \chi \rangle \rightarrow \langle \texttt{a1' / a2}, \sigma, \chi \rangle}$$ | Division — step left operand. |
| Rule 17 | $$\frac{\langle \texttt{a2}, \sigma, \chi \rangle \rightarrow \langle \texttt{a2'}, \sigma, \chi \rangle}{\langle \texttt{v1 / a2}, \sigma, \chi \rangle \rightarrow \langle \texttt{v1 / a2'}, \sigma, \chi \rangle}$$ | Division — step right operand. |
| Rule 18 | $$\frac{\texttt{v2} \neq 0 \quad \texttt{v3} = \texttt{v1}/\texttt{v2}}{\langle \texttt{v1 / v2}, \sigma, \chi \rangle \rightarrow \texttt{v3}}$$ | Division — compute (nonzero). |
| Rule 19 | $$\frac{\texttt{v2} = 0}{\langle \texttt{v1 / v2}, \sigma, \chi \rangle \rightarrow \langle \text{ERROR}, \sigma, \chi \rangle}$$ | Division by zero errors. |
| Rule 20 | $$\frac{\langle \texttt{a1}, \sigma, \chi \rangle \rightarrow \langle \texttt{a1'}, \sigma, \chi \rangle}{\langle \texttt{a1 \% a2}, \sigma, \chi \rangle \rightarrow \langle \texttt{a1' \% a2}, \sigma, \chi \rangle}$$ | Modulus — step left operand. |
| Rule 21 | $$\frac{\langle \texttt{a2}, \sigma, \chi \rangle \rightarrow \langle \texttt{a2'}, \sigma, \chi \rangle}{\langle \texttt{v1 \% a2}, \sigma, \chi \rangle \rightarrow \langle \texttt{v1 \% a2'}, \sigma, \chi \rangle}$$ | Modulus — step right operand. |
| Rule 22 | $$\frac{\texttt{v2} \neq 0 \quad \texttt{v3} = \texttt{v1 \% v2}}{\langle \texttt{v1 \% v2}, \sigma, \chi \rangle \rightarrow \texttt{v3}}$$ | Modulus — compute (nonzero). |
| Rule 23 | $$\frac{\texttt{v2} = 0}{\langle \texttt{v1 \% v2}, \sigma, \chi \rangle \rightarrow \langle \text{ERROR}, \sigma, \chi \rangle}$$ | Modulus by zero errors. |
| Rule 24 | $$\frac{\langle \texttt{a}, \sigma, \chi \rangle \rightarrow \langle \texttt{a'}, \sigma, \chi \rangle}{\langle \texttt{- a}, \sigma, \chi \rangle \rightarrow \langle \texttt{- a'}, \sigma, \chi \rangle}$$ | Unary minus — step. |

| Rule 25 | $$\frac{\text{v2} = -\text{v1}}{\langle \text{-- v1}, \sigma, \chi \rangle \to \text{v2}}$$ | Unary minus — compute. |
|---|---|---|
| Rule 26 | $$\frac{\langle \text{a}, \sigma, \chi \rangle \to \langle \text{a'}, \sigma, \chi \rangle}{\langle \text{+ a}, \sigma, \chi \rangle \to \langle \text{+ a'}, \sigma, \chi \rangle}$$ | Unary plus — step. |
| Rule 27 | $$\frac{}{\langle \text{+ v}, \sigma, \chi \rangle \to \text{v}}$$ | Unary plus — no-op. |
| Rule 28 | $$\frac{\langle \text{a1}, \sigma, \chi \rangle \to \langle \text{a1'}, \sigma, \chi \rangle}{\langle \text{a1 < a2}, \sigma, \chi \rangle \to \langle \text{a1' < a2}, \sigma, \chi \rangle}$$ | Less-than — step left. |
| Rule 29 | $$\frac{\langle \text{a2}, \sigma, \chi \rangle \to \langle \text{a2'}, \sigma, \chi \rangle}{\langle \text{v1 < a2}, \sigma, \chi \rangle \to \langle \text{v1 < a2'}, \sigma, \chi \rangle}$$ | Less-than — step right. |
| Rule 30 | $$\frac{\text{v1} < \text{v2}}{\langle \text{v1 < v2}, \sigma, \chi \rangle \to \text{true}}$$ | Less-than true. |
| Rule 31 | $$\frac{\text{v1} \geq \text{v2}}{\langle \text{v1 < v2}, \sigma, \chi \rangle \to \text{false}}$$ | Less-than false. |
| Rule 32 | $$\frac{\langle \text{a1}, \sigma, \chi \rangle \to \langle \text{a1'}, \sigma, \chi \rangle}{\langle \text{a1 <= a2}, \sigma, \chi \rangle \to \langle \text{a1' <= a2}, \sigma, \chi \rangle}$$ | Less-than-equal — step left. |
| Rule 33 | $$\frac{\langle \text{a2}, \sigma, \chi \rangle \to \langle \text{a2'}, \sigma, \chi \rangle}{\langle \text{v1 <= a2}, \sigma, \chi \rangle \to \langle \text{v1 <= a2'}, \sigma, \chi \rangle}$$ | Less-than-equal — step right. |
| Rule 34 | $$\frac{\text{v1} \leq \text{v2}}{\langle \text{v1 <= v2}, \sigma, \chi \rangle \to \text{true}}$$ | Less-than-equal true. |
| Rule 35 | $$\frac{\text{v1} > \text{v2}}{\langle \text{v1 <= v2}, \sigma, \chi \rangle \to \text{false}}$$ | Less-than-equal false. |
| Rule 36 | $$\frac{\langle \text{a1}, \sigma, \chi \rangle \to \langle \text{a1'}, \sigma, \chi \rangle}{\langle \text{a1 > a2}, \sigma, \chi \rangle \to \langle \text{a1' > a2}, \sigma, \chi \rangle}$$ | Greater-than — step left. |
| Rule 37 | $$\frac{\langle \text{a2}, \sigma, \chi \rangle \to \langle \text{a2'}, \sigma, \chi \rangle}{\langle \text{v1 > a2}, \sigma, \chi \rangle \to \langle \text{v1 > a2'}, \sigma, \chi \rangle}$$ | Greater-than — step right. |
| Rule 38 | $$\frac{\text{v1} > \text{v2}}{\langle \text{v1 > v2}, \sigma, \chi \rangle \to \text{true}}$$ | Greater-than true. |
| Rule 39 | $$\frac{\text{v1} \leq \text{v2}}{\langle \text{v1 > v2}, \sigma, \chi \rangle \to \text{false}}$$ | Greater-than false. |

| Rule 40 | $$\frac{\langle \texttt{a1}, \sigma, \chi\rangle \to \langle \texttt{a1'}, \sigma, \chi\rangle}{\langle \texttt{a1 >= a2}, \sigma, \chi\rangle \to \langle \texttt{a1' >= a2}, \sigma, \chi\rangle}$$ | Greater-than-equal — step left. |
|---|---|---|
| Rule 41 | $$\frac{\langle \texttt{a2}, \sigma, \chi\rangle \to \langle \texttt{a2'}, \sigma, \chi\rangle}{\langle \texttt{v1 >= a2}, \sigma, \chi\rangle \to \langle \texttt{v1 >= a2'}, \sigma, \chi\rangle}$$ | Greater-than-equal — step right. |
| Rule 42 | $$\frac{\texttt{v1} \geq \texttt{v2}}{\langle \texttt{v1 >= v2}, \sigma, \chi\rangle \to \texttt{true}}$$ | Greater-than-equal true. |
| Rule 43 | $$\frac{\texttt{v1} < \texttt{v2}}{\langle \texttt{v1 >= v2}, \sigma, \chi\rangle \to \texttt{false}}$$ | Greater-than-equal false. |
| Rule 44 | $$\frac{\langle \texttt{a1}, \sigma, \chi\rangle \to \langle \texttt{a1'}, \sigma, \chi\rangle}{\langle \texttt{a1 == a2}, \sigma, \chi\rangle \to \langle \texttt{a1' == a2}, \sigma, \chi\rangle}$$ | Equality — step left. |
| Rule 45 | $$\frac{\langle \texttt{a2}, \sigma, \chi\rangle \to \langle \texttt{a2'}, \sigma, \chi\rangle}{\langle \texttt{v1 == a2}, \sigma, \chi\rangle \to \langle \texttt{v1 == a2'}, \sigma, \chi\rangle}$$ | Equality — step right. |
| Rule 46 | $$\frac{\texttt{v1} = \texttt{v2}}{\langle \texttt{v1 == v2}, \sigma, \chi\rangle \to \texttt{true}}$$ | Equality true. |
| Rule 47 | $$\frac{\texttt{v1} \neq \texttt{v2}}{\langle \texttt{v1 == v2}, \sigma, \chi\rangle \to \texttt{false}}$$ | Equality false. |
| Rule 48 | $$\frac{\langle \texttt{a1}, \sigma, \chi\rangle \to \langle \texttt{a1'}, \sigma, \chi\rangle}{\langle \texttt{a1 != a2}, \sigma, \chi\rangle \to \langle \texttt{a1' != a2}, \sigma, \chi\rangle}$$ | Not-equal — step left. |
| Rule 49 | $$\frac{\langle \texttt{a2}, \sigma, \chi\rangle \to \langle \texttt{a2'}, \sigma, \chi\rangle}{\langle \texttt{v1 != a2}, \sigma, \chi\rangle \to \langle \texttt{v1 != a2'}, \sigma, \chi\rangle}$$ | Not-equal — step right. |
| Rule 50 | $$\frac{\texttt{v1} \neq \texttt{v2}}{\langle \texttt{v1 != v2}, \sigma, \chi\rangle \to \texttt{true}}$$ | Not-equal true. |
| Rule 51 | $$\frac{\texttt{v1} = \texttt{v2}}{\langle \texttt{v1 != v2}, \sigma, \chi\rangle \to \texttt{false}}$$ | Not-equal false. |
| Rule 52 | $$\frac{\langle \texttt{b1}, \sigma, \chi\rangle \to \langle \texttt{b1'}, \sigma, \chi\rangle}{\langle \texttt{b1 \&\& b2}, \sigma, \chi\rangle \to \langle \texttt{b1' \&\& b2}, \sigma, \chi\rangle}$$ | AND — step left. |
| Rule 53 | $$\frac{\langle \texttt{b2}, \sigma, \chi\rangle \to \langle \texttt{b2'}, \sigma, \chi\rangle}{\langle \texttt{q1 \&\& b2}, \sigma, \chi\rangle \to \langle \texttt{q1 \&\& b2'}, \sigma, \chi\rangle}$$ | AND — step right. |
| Rule 54 | $$\frac{\texttt{q1} = \texttt{true} \wedge \texttt{q2} = \texttt{true}}{\langle \texttt{q1 \&\& q2}, \sigma, \chi\rangle \to \texttt{true}}$$ | AND true. |

| Rule 55 | $$\frac{\text{q1} = \text{false} \lor \text{q2} = \text{false}}{\langle \text{q1 \&\& q2}, \sigma, \chi \rangle \to \text{false}}$$ | AND false. |
|---|---|---|
| Rule 56 | $$\frac{\langle \text{b1}, \sigma, \chi \rangle \to \langle \text{b1'}, \sigma, \chi \rangle}{\langle \text{b1 || b2}, \sigma, \chi \rangle \to \langle \text{b1' || b2}, \sigma, \chi \rangle}$$ | OR — step left. |
| Rule 57 | $$\frac{\langle \text{b2}, \sigma, \chi \rangle \to \langle \text{b2'}, \sigma, \chi \rangle}{\langle \text{q1 || b2}, \sigma, \chi \rangle \to \langle \text{q1 || b2'}, \sigma, \chi \rangle}$$ | OR — step right. |
| Rule 58 | $$\frac{\text{q1} = \text{true} \lor \text{q2} = \text{true}}{\langle \text{q1 || q2}, \sigma, \chi \rangle \to \text{true}}$$ | OR true. |
| Rule 59 | $$\frac{\text{q1} = \text{false} \land \text{q2} = \text{false}}{\langle \text{q1 || q2}, \sigma, \chi \rangle \to \text{false}}$$ | OR false. |
| Rule 60 | $$\frac{\langle \text{b}, \sigma, \chi \rangle \to \langle \text{b'}, \sigma, \chi \rangle}{\langle \text{!b}, \sigma, \chi \rangle \to \langle \text{!b'}, \sigma, \chi \rangle}$$ | NOT — step. |
| Rule 61 | $$\frac{\text{q} = \text{false}}{\langle \text{!q}, \sigma, \chi \rangle \to \text{true}}$$ | NOT of false is true. |
| Rule 62 | $$\frac{\text{q} = \text{true}}{\langle \text{!q}, \sigma, \chi \rangle \to \text{false}}$$ | NOT of true is false. |
| Rule 63 | $$\frac{\langle \text{s}, \sigma, \chi \rangle \to \langle \text{s'}, \sigma', \chi' \rangle}{\langle \text{s :: SL}, \sigma, \chi \rangle \to \langle \text{s' :: SL}, \sigma', \chi' \rangle}$$ | Sequence head steps. |
| Rule 64 | $$\frac{\langle \text{b}, \sigma, \chi \rangle \to \langle \text{b'}, \sigma, \chi \rangle}{\langle \text{if(b) \{SL1\} else \{SL2\} :: SL3}, \sigma, \chi \rangle \to \langle \text{if(b') \{SL1\} else \{SL2\} :: SL3}, \sigma, \chi \rangle}$$ | If-else predicate steps. |
| Rule 65 | $$\frac{\text{q} = \text{true}}{\langle \text{if(q) \{SL1\} else \{SL2\} :: SL3}, \sigma, \chi \rangle \to \langle \text{SL1 ++ SL3}, \sigma, \chi \rangle}$$ | If-else takes then-branch. |
| Rule 66 | $$\frac{\text{q} = \text{false}}{\langle \text{if(q) \{SL1\} else \{SL2\} :: SL3}, \sigma, \chi \rangle \to \langle \text{SL2 ++ SL3}, \sigma, \chi \rangle}$$ | If-else takes else-branch. |
| Rule 67 | $$\overline{\langle \text{while(b) \{SL\} :: SL1}, \sigma, \chi \rangle \to \langle \text{loop(b) \{SL\} :: SL1}, \sigma, \text{push}(\text{while(b) \{SL\}}, \chi) \rangle}$$ | While creates loop frame. |
| Rule 68 | $$\frac{\langle \text{b}, \sigma, \chi \rangle \to \langle \text{b'}, \sigma, \chi \rangle}{\langle \text{loop(b) \{SL\} :: SL1}, \sigma, \chi \rangle \to \langle \text{loop(b') \{SL\} :: SL1}, \sigma, \chi \rangle}$$ | Loop predicate steps. |
| Rule 69 | $$\frac{\text{q} = \text{false}}{\langle \text{loop(q) \{SL\} :: SL1}, \sigma, \chi \rangle \to \langle \text{SL1}, \sigma, \text{pop}(\chi) \rangle}$$ | Loop exits on false. |

| Rule 70 | $$\frac{q = \texttt{true}}{\langle \texttt{loop(q) \{SL\} ::\ SL1}, \sigma, \chi \rangle \rightarrow \langle \texttt{SL ++ (LE ::\ SL1)}, \sigma, \chi \rangle}$$ | Insert loop-body into statement list while adding a loop-end (LE) marker in between. |
|---|---|---|
| Rule 71 | $$\frac{\chi \neq \epsilon \ \wedge \ \texttt{s} \neq \texttt{LE}}{\langle \texttt{break ::\ s ::\ SL}, \sigma, \chi \rangle \rightarrow \langle \texttt{break ::\ SL}, \sigma, \chi \rangle}$$ | break propagates to LE inside loop. |
| Rule 72 | $$\frac{\chi \neq \epsilon \ \wedge \ \texttt{s} = \texttt{LE}}{\langle \texttt{break ::\ s ::\ SL}, \sigma, \chi \rangle \rightarrow \langle \texttt{SL}, \sigma, \text{pop}(\chi) \rangle}$$ | break at LE pops $\chi$ and terminates loop. |
| Rule 73 | $$\frac{\chi = \epsilon}{\langle \texttt{break ::\ SL}, \sigma, \chi \rangle \rightarrow \langle \texttt{ERROR}, \sigma, \chi \rangle}$$ | break outside loop errors. |
| Rule 74 | $$\frac{\chi \neq \epsilon \ \wedge \ \texttt{s} \neq \texttt{LE}}{\langle \texttt{continue ::\ s ::\ SL}, \sigma, \chi \rangle \rightarrow \langle \texttt{continue ::\ SL}, \sigma, \chi \rangle}$$ | continue propagates to LE inside loop. |
| Rule 75 | $$\frac{\chi \neq \epsilon \ \wedge \ \texttt{s} = \texttt{LE} \quad \texttt{s1} = \text{top}(\chi)}{\langle \texttt{continue ::\ s ::\ SL}, \sigma, \chi \rangle \rightarrow \langle \texttt{s1 ::\ SL}, \sigma, \text{pop}(\chi) \rangle}$$ | continue at LE pops $\chi$ and restarts loop. |
| Rule 76 | $$\frac{\chi = \epsilon}{\langle \texttt{continue ::\ SL}, \sigma, \chi \rangle \rightarrow \langle \texttt{ERROR}, \sigma, \chi \rangle}$$ | continue outside loop errors. |
| Rule 77 | $$\frac{\texttt{s} = \text{top}(\chi)}{\langle \texttt{LE ::\ SL}, \sigma, \chi \rangle \rightarrow \langle \texttt{s ::\ SL}, \sigma, \text{pop}(\chi) \rangle}$$ | LE pops $\chi$ and restarts loop. |
| Rule 78 | $$\frac{}{\langle \texttt{halt ::\ SL}, \sigma, \chi \rangle \rightarrow \langle \texttt{halt}, \sigma, \chi \rangle}$$ | Halt statement terminates program execution. |

## B IMP PROGRAM EXAMPLE

In this section we describe the collection of IMP programs for: (1) the Human-Written, (2) the LLM-Translated, (3) and the Fuzzer-Generated datasets and provide examples.

### B.1 HUMAN-WRITTEN DATASET

```
1  int sumEven(int l, int r)
2  {
3      int sum = 0;
4      for (int i = l; i <= r; i++)
5      {
6          if (i % 2 == 0)
7          {
8              sum += i;
9          }
10     }
11     return sum;
12 }
```

**(a)** The C++ solution to the problem "MBCPP/962" in BabelCode MBPP and one public test case. The public test we use is sumEven(3, 8)==18.

```
1  int sum;
2  int i;
3  int l;
4  int r;
5  l = 3;
6  r = 8;
7  i = l;
8  while(i <= r)
9  {
10     if((i % 2) == 0)
11     {
12         sum = (sum + i);
13     }
14     else
15     {
16
17     };
18     i = (i + 1);
19 };
```

**(b)** The IMP program (mbpp_962.imp in the Human-Written dataset) re-written from the C++ solution.

**Figure 4:** An example of re-writing a C++ program into an IMP program in the Human-Written dataset.

In Figure 4, we show an example C++ solution to a problem from the BabelCode MBPP benchmark (Figure 4a) and its corresponding IMP program re-written by us (Figure 4b). To convert the C++ program into an IMP program, we remove the function definitions (e.g., sumEven), while keeping the body of the function. Unsupported syntactic constructs are either re-written (e.g., replacing the for loop with a while loop) or removed (e.g., removing the return statement). One public test case is adopted as the program input and its output is used to verify correctness. In this example, l is assigned to 3 and r is assigned to 8, the test oracle 18 is used to verify the final-state of sum after program execution.

The code-complexity profile of the IMP program in Figure 4b is: control-flow complexity ($\Omega_{CC}$ = 3, $\Omega_{If}$ = 1, $\Omega_{Loop}$ = 1, $\hat{\Omega}_{If}$ = 1, $\hat{\Omega}_{Loop}$ = 1), data-flow complexity ($\Omega_{DD}$ = 12, $\hat{\Omega}_{Assign}$ = 12), and program-size complexity ($\Omega_{Loc}$ = 19, $\Omega_{Vol}$ = 294, $\Omega_{Voc}$ = 23, $\hat{\Omega}_{Trace}$ = 29).

## B.2 Fuzzer-Generated Dataset

The Fuzzer-Generated dataset is constructed using a semantic aware grammar based fuzzer with knobs for: (1) the generation probabilities of different statements, (2) the maximum nesting depth of the program (nested loops and conditionals), (3) the maximum and the minimum number of statements to generate per block, (4) the maximum number of terms and variable terms in arithmetic expressions, (5) the maximum number of terms in boolean expressions (relational and logical), and (6) the maximum and the minimum number of variable declarations in a program. We use the settings as shown in Table 11.

**Table 11:** Settings for the fuzzer knobs used to generate IMP programs for the Fuzzer-Generated dataset.

| Knob | Value |
|------|-------|
| **Structural limits** | |
| Minimum number of statements per block | 1 |
| Maximum number of statements per block | 3 |
| Minimum block depth | 5 |
| Maximum block depth | 10 |
| Minimum number of variables | 5 |
| Maximum number of variables | 10 |
| **Statement generation probabilities** | |
| Assignment | 0.4 |
| While | 0.3 |
| If | 0.2 |
| Break | 0.09 |
| Continue | 0.005 |
| Halt | 0.005 |
| **Expression limits** | |
| Maximum number of terms in arithmetic expr | 6 |
| Maximum number of variable terms in arithmetic expr | 3 |
| Maximum number of terms in boolean expr | 4 |

The fuzzer starts by randomly sampling an integer from the range defined by the minimum and maximum number of variable declarations. This integer specifies the number of variables to be declared and used for the IMP program being generated. The fuzzer next samples alphabets from the set {a-z} and {A-Z} until the required number of unique alphabets to use as variables is obtained. Declaration statments are then generated to declare these variables. Following this, one assignment statement is generated per declared variable to assign it with a randomly generated arithmetic expression. The arithmetic expression itself is generated using the pool of declared variables and integer constants (sampled from the set {0-9}).

The fuzzer next generates statements from the set {Assignment, While, If, Break, Continue, Halt} in accordance with the statement probabilities given in Table 11. No more than three statements are generated per block. These probabilities are used until the generation block depth reaches the specified minimum block depth (5). Beyond this, the statement probabilities are cosine-tapered to decrease the probabilities of generating `while` and `if-else` statements. For generation processes where the block depth reaches the maximum specified block depth (10), the probabilities of further generating `while` and `if-else` is reduced to zero.

To ensure high probability in termination of loops, the fuzzer generates one new variable (prefixed with `ble`) per loop. A monotone update type (incrementing or decrementing) is chosen for this variable each with a 50% probability of being chosen. The bounds, initial (before iteration) and expected final (after loop termination) values are then chosen from the range [-20,20] and the size of the update per iteration from the range [1 step, (final / 3) step]. The variable monotone update statement is inserted towards the end of the loop body and the bound is conjoined with the loop predicate. This prevents infinite loops. The declaration and assignment statements for these new generated variables is inserted right after the assignment statements for the intially chosen variables.

The fuzzer can be used to generate extremely complex IMP programs (as measured by the code-complexity metrics introduced earlier) with high probability of normal program termination. Figure 5 shows an example IMP program (`fuzz_100.imp`) from the Fuzzer-Generated dataset that was generated using our fuzzer. Its code-complexity metric profile is: control-flow complexity ($\Omega_{CC} = 62$, $\Omega_{If} = 5$, $\Omega_{Loop} = 6$, $\hat{\Omega}_{If} = 3$, $\hat{\Omega}_{Loop} = 5$), data-flow complexity ($\Omega_{DD} = 2603$, $\hat{\Omega}_{Assign} = 86$), and program-size complexity ($\Omega_{Loc} = 492$, $\Omega_{Vol} = 37140$, $\Omega_{Voc} = 91$, $\hat{\Omega}_{Trace} = 249$). This shows that out of the maximum loop nesting depth six ($\Omega_{Loop}$) present in the program, the execution reaches a maximum loop nesting depth of five ($\hat{\Omega}_{Loop}$) implying that the execution reached a loop contining four outer loops.

This is one of the programs from the Fuzzer-Generated dataset that the GEMINI-2.5-PRO model succesfully predicted the final-state of in the final-state prediction task.

```
1  int L;
2  int p;
```

```
3   int y;
4   int d;
5   int K;
6   int h;
7   int T;
8   int Y;
9   int ble0;
10  int ble1;
11  int ble2;
12  int ble3;
13  int ble4;
14  int ble5;
15  int ble6;
16  int ble7;
17  int ble8;
18  int ble9;
19  int ble10;
20  int ble11;
21  int ble12;
22  int ble13;
23  int ble14;
24  int ble15;
25  int ble16;
26  int ble17;
27  int ble18;
28  int ble19;
29  int ble20;
30  int ble21;
31  int ble22;
32  int ble23;
33  int ble24;
34  int ble25;
35  int ble26;
36  int ble27;
37  int ble28;
38  int ble29;
39  int ble30;
40  int ble31;
41  int ble32;
42  int ble33;
43  int ble34;
44  int ble35;
45  int ble36;
46  int ble37;
47  int ble38;
48  L = (((- y) / 4) - p);
49  T = ((((3 + K) + 1) - 3) + (8 / 8));
50  L = ((((((- p) % 1) * 7) - (- 1)) - (- 9)) - 3);
51  K = (((((d * (- 5)) + y) + (- 5)) - (- K)) - (- 5));
52  Y = (((9 + 3) - T) + 5);
53  K = ((7 / 7) * L);
54  y = ((((L + L) + 8) + 3) + 1);
55  p = ((((- 8) * 9) - ((- 6) % (- 8))) - T);
56  ble0 = (- 1);
57  ble1 = (- 1);
58  ble2 = (- 1);
59  ble3 = (- 1);
60  ble4 = (- 1);
61  ble5 = (- 1);
62  ble6 = (- 1);
63  ble7 = (- 1);
64  ble8 = (- 1);
65  ble9 = (- 1);
66  ble10 = (- 1);
67  ble11 = (- 1);
68  ble12 = (- 1);
69  ble13 = (- 1);
70  ble14 = (- 1);
71  ble15 = (- 1);
72  ble16 = (- 1);
73  ble17 = (- 1);
74  ble18 = (- 1);
75  ble19 = (- 1);
76  ble20 = (- 1);
77  ble21 = (- 1);
78  ble22 = (- 1);
79  ble23 = (- 1);
80  ble24 = (- 1);
81  ble25 = (- 1);
82  ble26 = (- 1);
83  ble27 = (- 1);
```

```
84   ble28 = (- 1);
85   ble29 = (- 1);
86   ble30 = (- 1);
87   ble31 = (- 1);
88   ble32 = (- 1);
89   ble33 = (- 1);
90   ble34 = (- 1);
91   ble35 = (- 1);
92   ble36 = (- 1);
93   ble37 = (- 1);
94   ble38 = (- 1);
95   if(((((y - K) - 2) == ((y % 8) % 5)) || ((L + y) < ((0 / 1) * (- K)))))
96   {
97      while(((((7 % 6) % 2) > (h + (- d))) || ((T % 1) != ((T * d) * (- 3)))) && (ble0 < 0))
98      {
99         while((((y + (- K)) <= (((- 1) + p) + L)) || (((p - (- L)) - 9) > (T - p))) && (ble1
               <= 5))
100        {
101           h = ((h - (4 % 2)) + (h * K));
102           ble1 = (ble1 + 2);
103        };
104        while(((((h - p) + 2) > ((9 * h) % 3)) || (((K + 4) - L) <= ((0 - h) + Y))) && (ble2
               < 20))
105        {
106           while(((!(((0 - d) + Y) != (h + L))) || ((d + (- d)) >= ((p - 1) - p))) && (ble3
                  < 15))
107           {
108              T = (((((1 + (- Y)) - 0) + h) - 4) - 1);
109              ble3 = (ble3 + 5);
110           };
111           Y = (1 + (5 / 6));
112           while((((!((d - L) <= ((1 * p) + L))) || ((h - K) >= ((9 - d) - (- h)))) && (ble4
                  < 9))
113           {
114              if((!((((- T) - ((- Y) % 8)) == (L + T))) || (((4 - y) + p) > (p * Y)))
115              {
116                 T = ((9 - Y) + (p % 1));
117                 while(((((- L) + y) <= ((6 * h) - K)) || ((1 + (Y % 9)) != ((p * y) + (- 7)
                        ))) && (ble5 <= 17))
118                 {
119                    K = ((9 + 9) + (5 * 9));
120                    ble5 = (ble5 + 3);
121                 };
122                 T = (((5 - 5) - 1) - (3 * 1));
123              }
124              else
125              {
126                 p = ((7 / (- 3)) + 8);
127                 break;
128              };
129              if(((d - L) < ((5 % 9) % 1)) && (!((T * K) <= (Y - K))))
130              {
131                 while((((T / 4) > ((- Y) - T)) && (((- 4) - ((- y) * (- T))) < (K + (3 / 3)
                        ))) && (ble6 < 13))
132                 {
133                    Y = (7 + (Y / 9));
134                    ble6 = (ble6 + 1);
135                 };
136                 if((((d + h) - 2) <= ((L - T) + 8)) && (((7 + (- h)) - h) == (L + T)))
137                 {
138                    while(((!(((((- y) - p) + 7) <= (d + d))) && ((Y + y) > (Y + h))) && (
                           ble7 > (- 12)))
139                    {
140                       break;
141                       ble7 = (ble7 + (- 3));
142                    };
143                 }
144                 else
145                 {
146                    h = ((8 - ((- 0) / 4)) + 2);
147                 };
148                 d = (((2 + (Y % 4)) - 8) - K);
149              }
150              else
151              {
152                 d = (((Y + Y) - L) - 2);
153              };
154              while((((T / (- 6)) < (p % 6)) || (((T + d) - 9) == ((9 + K) + h))) && (ble8 >
                     (- 20)))
155              {
```

```
156              while((((T - d) > (T + p)) && (((h - 9) + p) == ((p + 1) - p))) && (ble9 >
                     (- 15)))
157              {
158                  p = (((8 / (- 6)) + 0) + (4 / 8));
159                  K = ((((d % 5) + 7) + (9 / 2)) - 7);
160                  ble9 = (ble9 + (- 4));
161              };
162              ble8 = (ble8 + (- 1));
163          };
164          ble4 = (ble4 + 3);
165      };
166      ble2 = (ble2 + 6);
167  };
168  ble0 = (ble0 + 2);
169  };
170  }
171  else
172  {
173      p = (((L - y) - 4) + 5);
174      while((((p + p) <= (d + h)) && (((L - L) - (- 1)) <= (((- h) + 7) - p))) && (ble10 <=
            20))
175      {
176          while((((y / 7) >= ((5 - p) + (- Y))) || ((Y + (- Y)) >= ((4 * d) + (- Y)))) && (
                ble11 >= (- 9)))
177          {
178              if(((d - y) >= (h - K)) && ((T * (- T)) != ((- T) + (- Y))))
179              {
180                  break;
181                  break;
182                  if((((- Y) % 9) > (p + y)) && (!(((3 % 6) + y) != (L + K))))
183                  {
184                      break;
185                      K = (6 - ((- 6) % 5));
186                      if((((K + 6) + L) <= ((4 / 8) + Y)) || ((T * T) > (y + K)))
187                      {
188                          L = (d + (3 * 6));
189                          while(((((T - L) + 2) > (Y + T)) || (!((K + h) <= ((d + y) - 5)))) && (
                                ble12 < 11))
190                          {
191                              y = ((Y * h) - ((3 * 8) / 8));
192                              break;
193                              while(((((3 + d) - y) != (p / (- 6))) && (((- T) - h) >= (L / 1))) &&
                                    (ble13 >= (- 17)))
194                              {
195                                  L = (((Y / 2) * T) + (8 % (- 9)));
196                                  y = ((- y) + ((- p) * T));
197                                  p = (((- 6) + (8 * 5)) - 8);
198                                  ble13 = (ble13 + (- 3));
199                              };
200                              ble12 = (ble12 + 3);
201                          };
202                          p = ((((K * (- 6)) * 3) - 2) + 7);
203                      }
204                      else
205                      {
206                          Y = (((8 % 4) - (p * 4)) - 8);
207                          if((((d + (- d)) + 0) == (((- d) / 4) * K)) || ((Y * y) >= ((1 % 8) + (-
                                L))))
208                          {
209                              y = (((5 * p) + T) - d);
210                              p = ((0 * 0) - (((0 / 3) % 1) / 9));
211                          }
212                          else
213                          {
214                              while((((K - (d * 2)) != (p / 2)) || (((L / 9) - y) < (Y - T))) && (
                                    ble14 < 20))
215                              {
216                                  y = ((p - (0 * (- h))) + L);
217                                  p = (((((- 4) - 6) - y) + L) + T);
218                                  break;
219                                  ble14 = (ble14 + 1);
220                              };
221                              h = ((T - 4) + 9);
222                              T = (((((- 5) - 3) + 2) - 1) + 8);
223                          };
224                          y = ((((2 + (9 / 9)) + 4) - (- 0)) + 1);
225                      };
226                  }
227                  else
228                  {
229                      break;
```

```
1404
1405    230                          };
        231                      }
1406    232                  else
        233                  {
1407    234                      while((((((- h) + 4) + K) <= (h - Y)) || (((6 * p) + d) < (T / (- 2)))) && (
1408                                  ble15 < 9))
        235                      {
1409    236                          L = (Y - (((d * h) / (- 8)) % 1));
        237                          K = (((((- 4) * 9) + (7 * 2)) + 1) + 1);
1410    238                          d = (((6 - (L * 5)) - 0) + 8);
        239                          ble15 = (ble15 + 3);
1411    240                      };
        241                      while(((!(((((- 1) * p) - y) != (y - h))) || ((((- 4) / (- 5)) + d) <= (y + (-
1413                                  h)))) && (ble16 > (- 7)))
1414    242                      {
        243                          break;
1415    244                          K = (((d * h) + 5) - 7);
        245                          while(((((d % (- 1)) <= ((- p) * K)) && ((h - y) == (p - h))) && (ble17 > (-
1416                                      13)))
1417    246                          {
        247                              T = (((((0 - T) + 0) + 6) + 2) - 2);
1418    248                              break;
        249                              break;
1419    250                              ble17 = (ble17 + (- 3));
1420    251                          };
        252                          ble16 = (ble16 + (- 2));
1421    253                      };
        254                      while(((((Y + (T * 3)) == (((- d) - (- 6)) + p)) || (((L % 8) - L) == (T + h)))
1423                                  && (ble18 >= (- 12)))
1424    255                      {
        256                          d = (((6 % (- 6)) - (K * T)) + L);
1425    257                          if((((5 * d) + h) > ((L * 4) / 9)) && ((K + L) > (Y - y)))
        258                          {
1426    259                              K = ((8 + ((7 * 1) / 7)) - (- 3));
        260                              d = ((p - ((- 2) / 2)) - (1 * 6));
1428    261                          }
        262                          else
1429    263                          {
        264                              break;
1430    265                          };
        266                          if((!((0 - (d % 5)) != (L * d))) || ((d + p) >= ((8 + d) + (- K))))
1431    267                          {
        268                              d = (((K / 8) % 6) - (T % (- 9)));
1433    269                              if(((h / 6) >= ((- K) / 1)) || (((h - d) + (- 7)) >= ((y + 1) - h)))
1434    270                              {
        271                                  Y = ((((T - (p * T)) - 2) - 4) + 4);
1435    272                              }
        273                              else
1436    274                              {
        275                                  K = (6 + ((- 0) % 2));
1437    276                                  h = (((8 + T) + 8) - 2);
        277                              };
1438    278                          }
        279                          else
1439    280                          {
        281                              if((((((- 3) + h) - p) < ((9 + d) + L)) && ((K + d) > (d * y)))
1441    282                              {
        283                                  if(((h / 6) < ((y - d) - 4)) || ((h + L) != (y - (T / 3))))
1443    284                                  {
        285                                      Y = ((- 8) + ((- 5) % 7));
1444    286                                  }
        287                                  else
1445    288                                  {
        289                                      d = (((6 + 8) + 4) - (- 6));
1447    290                                  };
        291                                  while((((p + (T % 4)) <= ((6 + L) - p)) || (((0 * p) - (- K)) >= (((-
1448                                              1) + (- K)) + Y))) && (ble19 < 12))
1449    292                                  {
        293                                      break;
1450    294                                      T = (Y + (T / 5));
        295                                      ble19 = (ble19 + 1);
1451    296                                  };
        297                                  Y = ((L - (5 % 8)) + ((T % 1) % 2));
1452    298                              }
        299                              else
1454    300                              {
        301                                  while((((p * d) != (((- L) + T) + 9)) && (!(((8 + d) - y) < (p + y)))
1456                                              ) && (ble20 <= 18))
        302                                  {
1457    303                                      if(((d - (3 * L)) < ((p + d) - 6)) || ((T - d) <= ((T % 1) + (- L)
                                                )))
```

```
304                                    {
305                                        break;
306                                    }
307                                    else
308                                    {
309                                        break;
310                                        h = (((( 5 + 9) - 1) - (4 / 3)) - 3);
311                                        L = (((p + (0 * 8)) + 0) + 8);
312                                    };
313                                    ble20 = (ble20 + 5);
314                                };
315                            };
316                        };
317                        ble18 = (ble18 + (- 3));
318                    };
319                };
320                if(((Y * (- K)) == (T - h)) || (((L % 6) - (- K)) >= ((K / (- 2)) / 6)))
321                {
322                    h = ((((4 % 6) + (6 % 1)) + 3) + (- 1));
323                    while((((K + y) == ((- T) - (- L))) && ((Y / 9) != (((- p) * h) + 7))) && (
                            ble21 >= (- 8)))
324                    {
325                        if((!((((4 % 6) % 6) >= (p / 8))) && (((2 + y) - K) >= ((1 + y) - (- y))))
326                        {
327                            T = (((((- L) * 1) - (5 * (- 8))) + 6);
328                        }
329                        else
330                        {
331                            h = (((((0 - y) + L) + 5) + T) + 5);
332                        };
333                        if(((Y * p) <= ((4 % 2) + T)) && ((d + L) >= ((y - (- 1)) + h)))
334                        {
335                            h = ((((6 + 2) - (- Y)) + y) - 1);
336                        }
337                        else
338                        {
339                            d = ((((L * (- 5)) - T) - (- L)) - (2 / 8));
340                            while(((((K + (- K)) + 8) != (d + h)) || ((Y * (- K)) < (T - (- K)))) &&
                                    (ble22 > (- 20)))
341                            {
342                                while((((y % 3) >= ((6 - (- Y)) + T)) && ((y + (- K)) >= ((K + 8) + p
                                        ))) && (ble23 < 18))
343                                {
344                                    K = (((4 % (- 9)) + (- K)) + y);
345                                    h = ((2 * 7) - (7 % 7));
346                                    break;
347                                    ble23 = (ble23 + 1);
348                                };
349                                y = (((T - 8) + ((9 % 3) % 1)) + (- 8));
350                                ble22 = (ble22 + (- 2));
351                            };
352                        };
353                        y = ((T + 2) - ((((- y) / 9) % 6) / (- 9)));
354                        ble21 = (ble21 + (- 3));
355                    };
356                    y = ((d - 4) - 7);
357                }
358                else
359                {
360                    while((((y * K) != (L + (- Y))) || (((3 / 2) - p) > (K / 4))) && (ble24 >= (-
                            4)))
361                    {
362                        while((((d + Y) < (h - L)) || (((Y + (- h)) + 7) >= (T - d))) && (ble25 <=
                                5))
363                        {
364                            while((((K - T) <= (T - (- Y))) || ((y + d) >= ((p % 9) * T))) && (ble26
                                    > (- 15)))
365                            {
366                                break;
367                                ble26 = (ble26 + (- 2));
368                            };
369                            ble25 = (ble25 + 2);
370                        };
371                        d = (((((- L) - h) + K) + (3 * 2)) + 7);
372                        while(((!((((T - L) + (- 5)) >= ((2 * (- K)) + T))) || (((8 + (- T)) - (- y)
                                ) <= (y / 4))) && (ble27 > (- 19)))
373                        {
374                            while((((8 + (L * y)) >= (d / 8)) || (((y / 3) + T) < ((7 * h) + (- Y)))
                                    ) && (ble28 > (- 9)))
375                            {
376                                break;
```

```
377                         while(((((Y + (Y * (- 5))) >= ((p * (- h)) - 4)) && ((p % 3) > (Y - h)
                                )) && (ble29 >= (- 10)))
378                         {
379                             break;
380                             ble29 = (ble29 + (- 2));
381                         };
382                         ble28 = (ble28 + (- 1));
383                     };
384                     K = (((((p + d) + 0) + (- 9)) - 0) + 9);
385                     if(((((h * 2) + T) == (y - h)) || ((y % 7) < (L + p)))
386                     {
387                         while(((((y - (Y % 5)) == ((L * (- L)) + 0)) || (((- 7) + (p * T)) > (
                                L / 5))) && (ble30 >= (- 2)))
388                         {
389                             h = ((3 + (- d)) - ((- T) * Y));
390                             T = (((((- 6) * L) - y) + 7) + 6);
391                             ble30 = (ble30 + (- 2));
392                         };
393                         if(((((h * 5) % 7) <= ((d - (- Y)) - 4)) && ((L + Y) <= ((T * 5) % 4))
                                )
394                         {
395                             Y = (((3 + 7) + 3) - 9);
396                             T = ((3 + L) - ((y % (- 6)) * (- h)));
397                             h = ((h * T) - (7 / 4));
398                         }
399                         else
400                         {
401                             Y = ((((0 - (- 8)) + 5) + 6) - (5 * 9));
402                             T = ((Y + (- 5)) + (- 6));
403                         };
404                     }
405                     else
406                     {
407                         Y = (((3 - K) - Y) + (0 / 1));
408                         while((((L - h) < (y + y)) && ((p * y) == (((- h) * K) % (- 2)))) &&
                                (ble31 <= 17))
409                         {
410                             K = (((y * y) - d) + 7);
411                             while((((L / 7) != ((- p) / 6)) && (!(((4 - p) - T) == ((5 * K) /
                                    (- 2))))) && (ble32 > (- 6)))
412                             {
413                                 L = ((((- K) + (- 4)) - p) + (- d));
414                                 d = ((y + 9) - 1);
415                                 h = ((((K / 4) - 8) - (- 4)) + ((- 6) * 5));
416                                 ble32 = (ble32 + (- 2));
417                             };
418                             K = (((6 + 6) - 6) + (p / 2));
419                             ble31 = (ble31 + 6);
420                         };
421                         while((((8 - (h * (- p))) != (((- Y) / 2) + d)) && ((h - T) - (- 6))
                                > (y * d))) && (ble33 <= 0))
422                         {
423                             T = (((((- 2) + 7) + (- 9)) + 0) + 0);
424                             break;
425                             ble33 = (ble33 + 2);
426                         };
427                     };
428                     ble27 = (ble27 + (- 5));
429                 };
430                 ble24 = (ble24 + (- 2));
431             };
432         };
433         while(((((L * 2) - T) != (d + L)) || (((2 + (- T)) - d) == ((6 - h) - L))) && (
                ble34 > (- 7)))
434         {
435             d = ((K * Y) + L);
436             L = ((4 % 8) % 4);
437             ble34 = (ble34 + (- 1));
438         };
439         ble11 = (ble11 + (- 3));
440     };
441     T = ((((- 3) * (- K)) + 4) - (- 7));
442     h = (((((7 + 4) + 4) - (- 9)) + p) - 2);
443     ble10 = (ble10 + 5);
444 };
445 if(((((Y * (- 5)) + L) > (((- 9) - Y) - T)) || (((L / 8) % 4) == (K - K)))
446 {
447     while(((!((h + T) != (K + ((- Y) * 7)))) || (!(((L / 9) - T) < ((Y + T) + (- 8)))))
            && (ble35 <= (- 3)))
448     {
449         Y = ((0 / 7) - 2);
```

```
450          Y = ((T + 2) + 8);
451          ble35 = (ble35 + 2);
452       };
453       while(((((3 + K) + L) != ((3 - h) + d)) && ((d + (- L)) > ((K - 8) - y))) && (ble36
              < (- 1)))
454       {
455          if(((K + T) > (d + K)) || ((y - K) == ((L + 1) + p)))
456          {
457             L = (((8 * p) * p) * L);
458          }
459          else
460          {
461             if((!(((( - K) * T) + 8) != (L + K))) || (((6 % 1) - p) != ((y + 1) - K)))
462             {
463                y = ((((Y - 0) + 5) - h) + (- L));
464                break;
465                while((((((- Y) - h) + 9) <= (Y * p)) || (((( - 6) - K) + Y) <= (Y - y))) &&
                        (ble37 < 20))
466                {
467                   K = ((p % 7) + ((d / 6) * (- 3)));
468                   while(((((y + (K * 7)) == ((- d) + T)) || ((h * h) >= (6 + (L * Y)))) &&
                           (ble38 > (- 18)))
469                   {
470                      Y = (3 - (L % 9));
471                      T = ((p + Y) + L);
472                      ble38 = (ble38 + (- 2));
473                   };
474                   K = ((p - Y) - (((- T) * 1) / 3));
475                   ble37 = (ble37 + 6);
476                };
477             }
478             else
479             {
480                h = ((9 + (- p)) + 8);
481                y = ((((K * 7) % 6) / 4) + T);
482             };
483          };
484          h = (((L + (2 * 6)) - 1) + (- 3));
485          ble36 = (ble36 + 2);
486       };
487    }
488    else
489    {
490       Y = (((6 - p) - (4 * (- Y))) - L);
491    };
492 };
```

**Figure 5:** An example IMP program (fuzz_100.imp) from the Fuzzer-Generated dataset. Its code-complexity metric profile is: control-flow complexity ($\Omega_{CC} = 62$, $\Omega_{If} = 5$, $\Omega_{Loop} = 6$, $\hat{\Omega}_{If} = 3$, $\hat{\Omega}_{Loop} = 5$), data-flow complexity ($\Omega_{DD} = 2603$, $\hat{\Omega}_{Assign} = 86$), and program-size complexity ($\Omega_{Loc} = 492$, $\Omega_{Vol} = 37140$, $\Omega_{Voc} = 91$, $\hat{\Omega}_{Trace} = 249$).

## C CODE-COMPLEXITY DISTRIBUTIONS

The distributions of the code-complexity metrics used to characterize the control-flow, data-flow, and the program size complexity are given in Figure 6. We mark the median and the extremas for each distribution. We see that the median $\Omega_{If}$ and $\Omega_{Loop}$ is similar for the Human-Written and the LLM-Translated datasets, whereas for every other metric, the LLM-Translated has slightly higher median values than Human-Written and thus more complex programs. The Fuzzer-Generated dataset on the other hand has median values significantly higher for every metric except $\hat{\Omega}_{Trace}$ and $\hat{\Omega}_{Assign}$, than the other two datasets. This implies that programs in the Fuzzer-Generated and the LLM-Translated datasets run for roughly the same number of execution steps (measured as per the SOS semantics) but the programs in the former are significantly more complex than those in the latter.

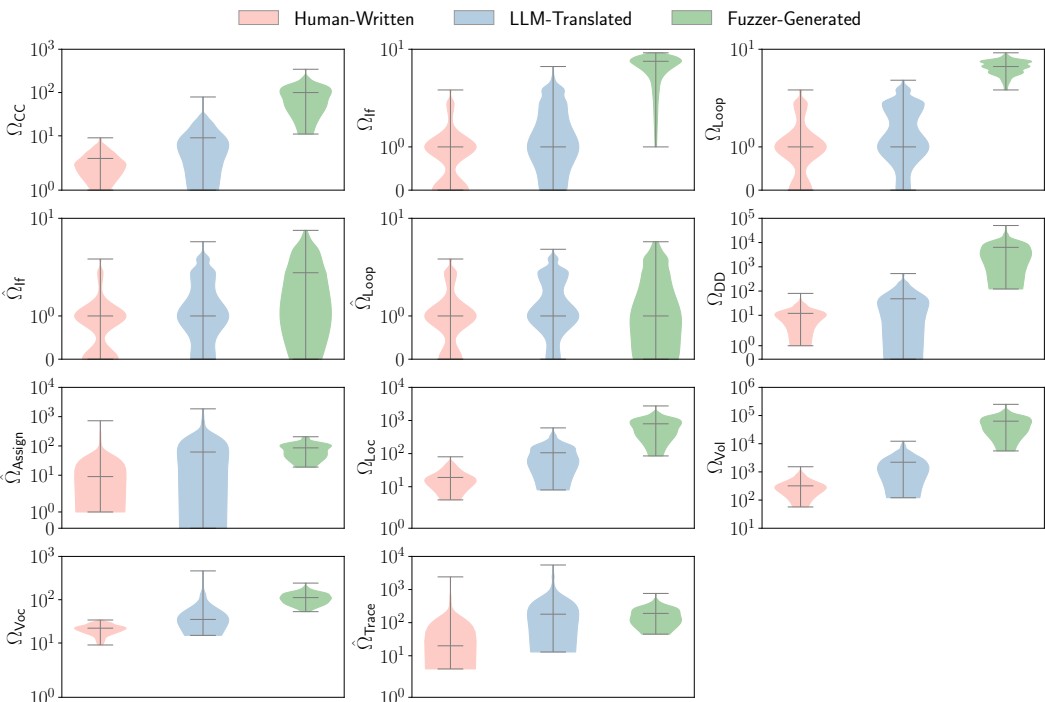

**Figure 6:** Distributions of the code-complexity metrics extended cyclomatic complexity ($\Omega_{CC}$), maximum nested if–else ($\Omega_{If}$) and nested loop ($\Omega_{Loop}$) depths , maximum taken nested if–else ($\hat{\Omega}_{If}$), and taken nested loop ($\hat{\Omega}_{Loop}$) depths, the program data-flow complexity metrics DepDegree ($\Omega_{DD}$) and the total number of assignments to variables in execution traces ($\hat{\Omega}_{Assign}$), and finally the program size complexity metrics, lines of code ($\Omega_{Loc}$), Halstead metrics Volume ($\Omega_{Vol}$) and Vocabulary($\Omega_{Voc}$), and execution trace length ($\hat{\Omega}_{Trace}$).

## D EXPERIMENTS DETAILS

### D.1 PARAMETERS

We use a temperature of 0.6 for DeepSeek distilled models and QWQ 32B for improved reasoning. We use the default temperature settings for O3-MINI,GPT-5-MINI, and GEMINI-2.5-PRO by not specifying a specific temperature. For other non-reasoning models, we set the temperature to zero. All models are evaluated under the zero-shot setting.

### D.2 COMPUTE RESOURCES

The experiments on open-weight models with fewer than 70 billion parameters are conducted on a single compute node equipped with one NVIDIA H200 GPU (96 GB memory), an NVIDIA

Grace CPU @ 3.1 GHz with 72 cores, and 116 GB LPDDR5 memory. For experiments involving 70B-parameter models, we use four compute nodes.

## D.3 PROMPTS

---

**D.3.1 Prompt for PredState task.**

**No-semantics**:
```
You are an interpreter for my language called {language}.

Here is the {language} program
     {program}
```

**SOS**:
```
You are an interpreter for a language called {language}.  I will
describe the syntax for {language} in EBNF and its semantics using
small-step operational semantics.  You will use this to execute a
{language} program.  You will only use the rules described in the
semantics I provide.  Assume all the rules in the semantics I give are
correct.  A program has finished execution when one of the terminal
configurations ⟨ϵ,σ,χ⟩,⟨{HALT},σ,χ⟩, ⟨{ERROR},σ,χ⟩ is reached.

Here is the syntax of {language} in EBNF
     {syntax}

Here is the small-step operational semantics of {language}
     {semantics}

Here is the {language} program
     {program}
```

**K-semantics**:
```
You are an interpreter for a language called {language}.  I will
describe the syntax and the semantics of the language using the
K-framework.  You will use this to execute a {language} program.  You
will only use the rules described in the semantics I provide.  Assume
all the rules in the semantics I give are correct.

Here is the K-framework formalization of {language}
     {semantics}

Here is the {language} program
     {program}

## TASK: predict the values of all the declared variables after
executing the above program.
- If you think the program will never terminate, answer with the
special word '##timeout##':

     <answer>##timeout##</answer>

- If you believe the program has an error or has undefined behavior,
answer with the special word '##error##':

     <answer>##error##</answer>

- Otherwise, provide the predicted values of all the declared variables
in the following format:

     <answer>[Your answer]</answer>

Here is one example:
```

---

```
** Program **
int a;
int b;
int ans;
int c;
a {ASSIGN_OP} 10;
b {ASSIGN_OP} 23;
c {ASSIGN_OP} 12;
ans {ASSIGN_OP} a {ADD_OP} b;

The final expected output is:
<answer>
  <a>10</a>
  23
  <c>12</c>
  <ans>33</ans>
</answer>
```

**Non-CoT:** Only write the answer. You **MUST** wrap your prediction with `<ans>` tags.
**CoT:** Explain your reasoning step-by-step **before** answering. Wrap your reasoning in `<reason>` tags. Note that you **MUST** wrap your reasoning steps with `<reason>` tags and the prediction with `<ans>` tags.

### D.3.2 Prompt for PredRule task.

**SOS:**
You are an interpreter for a language called {language}. I will describe the syntax for {language} in EBNF and its semantics using small-step operational semantics. You will use this to execute a {language} program. You will only use the rules described in the semantics I provide. Assume all the rules in the semantics I give are correct. A program has finished execution when one of the terminal configurations $\langle \epsilon, \sigma, \chi \rangle, \langle \{\text{HALT}\}, \sigma, \chi \rangle, \langle \{\text{ERROR}\}, \sigma, \chi \rangle$ is reached.

Here is the syntax of {language} in EBNF
    {syntax}

Here is the small-step operational semantics of {language}
    {semantics}

Here is the {language} program
    {program}

## TASK:
For each question below, you'll be given:
1. A program
2. The program state ($\sigma$) (variable values) before executing the program
3. The control stack ($\chi$) before executing the program

Assume that all necessary variables have been declared and have the values as indicated in the provided program state.
You must:
- Correctly identify and apply the small-step operational semantic rules required to evaluate the program to completion
- List them in the correct order of application

A program is executed completely when its evaluation reaches one of the terminal configurations $\langle \epsilon, \sigma, \chi \rangle, \langle \{\text{HALT}\}, \sigma, \chi \rangle, \langle \{\text{ERROR}\}, \sigma, \chi \rangle$.

```
Here is one example:
** Program:**
{WHILE} (n {LTEQ_OP} 0)
{{
    {HALT};
}};

**Program state(σ) before execution:**
{{'n':  100, 'sum':  0}}

**Control stack(χ) before execution:**
ϵ

This is the sequence of steps:
1.  First, we transform the {WHILE} into {LOOP} using **Rule 67**.
2.  Reduce the loop predicate using **Rule 68**.
3.  The loop predicate is a {LTEQ_OP} operator which triggers **Rule
32** to first reduce the left-hand side 'n' to a literal using **Rule
1**.
4.  The right-hand side is already a literal and since '100' is not
less-than or equal to '0'.  We use **Rule 35** to evaluate this
operation to 'false'.
5.  Since the loop predicate is 'false', we use **Rule 69** to
terminate the loop.
6.  Since there are no more statements left, we have reached the
terminal configuration ⟨ϵ,σ,χ⟩ and the program evaluation terminates.

Therefore, the final answer is:
<ans>
  <answer id="1">
    <rule>67</rule>
    <rule>68</rule>
    <rule>32</rule>
    <rule>1</rule>
    <rule>35</rule>
    <rule>69</rule>
  </answer>
</ans>

## Questions:
{questions}

## Response Format:
Respond with an XML block structured as follows:

<ans>
  <answer id="1">
    <rule>1</rule>
    <rule>2</rule>
  ...
  </answer>
  <answer id="2">
    <rule>1</rule>
    <rule>2</rule>
  ...
  </answer>
  ...
</ans>

### Notes:
```

```
- Each <answer id="N"> element corresponds to the N-th question.
- Inside each <answer> block, list each semantic rule in the correct
order using <rule> tags.

## Important Notes:
- The **order** of rules matters and should reflect the evaluation
sequence.
- A single rule may be needed to be applied multiple times during
evaluation.
- You must include **all** semantic rules required for complete
execution.
- Base your analysis solely on the provided semantics, not on general
programming knowledge.
```

**K-semantics :**
```
You are an interpreter for a language called {language}.  I will
describe the syntax and the semantics of the language using the
K-framework.  You will use this to execute a {language} program.  You
will only use the rules described in the semantics I provide.  Assume
all the rules in the semantics I give are correct.

Here is the K-framework formalization of {language}
      {semantics}

Here is the {language} program
      {program}

## TASK:
For each question below, you'll be given:
1.  A program
2.  The program state (σ) (variable values) before executing the
program
3.  The control stack (χ) before executing the program

Assume that all necessary variables have been declared and have the
values as
indicated in the provided program state.

You must:
- Correctly identify and apply the K-semantic rules required to
evaluate the program to completion
- List them in the correct order of application

Here is one example:
** Program:**
{WHILE} (n {LTEQ_OP} 0)
{{
    {HALT};
}};

**Program state(σ) before execution:**
{{'n':  100, 'sum':  0}}

**Control stack(χ) before execution:**
ε

This is the sequence of steps:
1.  First, we transform the '{WHILE}' into '{WHILE}1' while also
inserting a 'breakMarker' after '{WHILE}1' using **Rule 24**.
2.  Next we transform the '{WHILE}1' into an '{IF}-{ELSE}' with the
```

```
'{WHILE}1' as the body of the '{IF}' using **Rule 25**.
3.   We then reduce the loop predicate to a boolean by first reducing
left-hand-side which is a variable using **Rule 1** and then applying
the '{LTEQ_OP}' using **Rule 13*.
4.   Since the loop predicate evaluates to 'false', we apply the '{IF}'
not taken rule **Rule 23** to take the '{ELSE}' branch which is empty.
5.   Finally, we evaluate the 'breakMarker' statement using **Rule 27**
to conclude the program execution.

Therefore, the final answer is:
<ans>
  <answer id="1">
    <rule>24</rule>
    <rule>25</rule>
    <rule>1</rule>
    <rule>13</rule>
    <rule>23</rule>
    <rule>27</rule>
  </answer>
</ans>

## Questions:
{questions}

## Response Format:
Respond with an XML block structured as follows:

<ans>
  <answer id="1">
    <rule>1</rule>
    <rule>2</rule>
  ...
  </answer>
  <answer id="2">
    <rule>1</rule>
    <rule>2</rule>
  ...
  </answer>
  ...
</ans>

### Notes:
- Each '<answer id="N">' element corresponds to the N-th question.
- Inside each '<answer>' block, list each semantic rule in the correct
order using '<rule>' tags.

## Important Notes:
- The **order** of rules matters and should reflect the evaluation
sequence.
- Only rules that have names indicated in '[]' adjacent to it must be
reported in the answer.
- A single rule may be needed to be applied multiple times during
evaluation.
- You must include **all** semantic rules required for complete
execution.
- Base your analysis solely on the provided semantics, not on general
programming knowledge.

Non-CoT: Only output the '<ans>' XML block.  Do not include any other
content.
CoT:  Explain your reasoning step-by-step **before** answering.  Wrap
```

1944
1945
1946
1947
1948
1949
1950
1951
1952
1953
1954
1955
1956
1957
1958
1959
1960
1961
1962
1963
1964
1965
1966
1967
1968
1969
1970
1971
1972
1973
1974
1975
1976
1977
1978
1979
1980
1981
1982
1983
1984
1985
1986
1987
1988
1989
1990
1991
1992
1993
1994
1995
1996
1997

```
your reasoning in '<reason>' tags.
```

**D.3.3 Prompt for PredTrace task.**

**SOS**:
```
You are an interpreter for a language called {language}.  I will
describe the syntax for {language} in EBNF and its semantics using
small-step operational semantics.  You will use this to execute a
{language} program.  You will only use the rules described in the
semantics I provide.  Assume all the rules in the semantics I give are
correct.  A program has finished execution when one of the terminal
configurations ⟨ε,σ,χ⟩,⟨{HALT},σ,χ⟩,  ⟨{ERROR},σ,χ⟩ is reached.

Here is the syntax of {language} in EBNF
      {syntax}

Here is the small-step operational semantics of {language}
      {semantics}

Here is the {language} program
      {program}

## TASK:
Given a program and its semantics, predict the execution trace.  Your
goal is to simulate execution, step by step of executing the program
using the given small-step operational semantics rules.  Do not skip
any rules that is needed to evaluate the program.  You will output your
answer in the following format.

## Response Format:
Respond with an XML block structured as follows:

<answer>
  <step>
    <rule>1</rule>
    <program_state>
      <n>0</n>
      <sum>0</sum>
    </program_state>
  </step>
  <step>
    <rule>2</rule>
    <program_state>
      <n>100</n>
      <sum>0</sum>
    </program_state>
  </step>
...
</answer>

## Here is an example:

Here is the {language} program:
int i;
int j;
i {ASSIGN_OP} 0;
{WHILE} (i {LT_OP} 2)
{{      {HALT};
}};

## Expected output:
<answer>
```

```
<step>
  <rule>3</rule>
  <program_state>
    0
  </program_state>
</step>
<step>
  <rule>3</rule>
  <program_state>
    0
    <j>0</j>
  </program_state>
</step>
<step>
  <rule>5</rule>
  <program_state>
    0
    <j>0</j>
  </program_state>
</step>
<step>
  <rule>67</rule>
  <program_state>
    0
    <j>0</j>
  </program_state>
</step>
<step>
  <rule>68</rule>
  <program_state>
    0
    <j>0</j>
  </program_state>
</step>
<step>
  <rule>28</rule>
  <program_state>
    0
    <j>0</j>
  </program_state>
</step>
<step>
  <rule>1</rule>
  <program_state>
    0
    <j>0</j>
  </program_state>
</step>
<step>
  <rule>30</rule>
  <program_state>
    0
    <j>0</j>
  </program_state>
</step>
<step>
  <rule>70</rule>
  <program_state>
    0
    <j>0</j>
  </program_state>
</step>
<step>
  <rule>78</rule>
```

```
      <program_state>
        0
        <j>0</j>
      </program_state>
    </step>
</answer>

 ## Notes:
 - Each '<step>' must correspond to **exactly one small-step operational
 semantics rule** that is needed to evaluate a statement in the given
 program.
 - The '<rule>' must indicate a rule used in the evaluation of a
 statement.
 - The '<program_state>' must represent the **entire program state
 immediately after** the execution of that rule.
 - The program state must list **all variables currently in scope**,
 using the variable names as XML tags and their current values as tag
 content.
 - Include variables even if they did not change.
 - Do not skip any step or merge multiple steps into one.
 - Do not skip any rules (including those used to reduce expressions and
 variables) that are needed to evaluate the program.
 - The program execution is complete when on of the terminal
 configurations ⟨ϵ,σ,χ⟩,⟨{HALT},σ,χ⟩, ⟨{ERROR},σ,χ⟩ is reached
```

**K-semantics:**
```
You are an interpreter for a language called {language}.  I will
describe the syntax and the semantics of the language using the
K-framework.  You will use this to execute a {language} program.  You
will only use the rules described in the semantics I provide.  Assume
all the rules in the semantics I give are correct.

Here is the K-framework formalization of {language}
      {semantics}

Here is the {language} program
      {program}

 ## TASK:
 Given a program and its semantics, predict the execution trace.  Your
 goal is to simulate execution, step by step of executing the program
 using the given small-step operational semantics rules.  Do not skip
 any rules that is needed to evaluate the program.  You will output your
 answer in the following format.

 ## Response Format:
 Respond with an XML block structured as follows:

 <answer>
   <step>
     <rule>1</rule>
     <program_state>
       <n>0</n>
       <sum>0</sum>
     </program_state>
   </step>
   <step>
     <rule>2</rule>
     <program_state>
       <n>100</n>
       <sum>0</sum>
```

```
    </program_state>
  </step>
...
</answer>

## Here is an example:

Here is the {language} program:
int i;
int j;
i {ASSIGN_OP} 0;
{WHILE} (i {LT_OP} 2)
{{
    {HALT};
}};

## Expected output:

<answer>
  <step>
    <rule>36</rule>
    <program_state>
      0
    </program_state>
  </step>
  <step>
    <rule>36</rule>
    <program_state>
      0
      <j>0</j>
    </program_state>
  </step>
  <step>
    <rule>21</rule>
    <program_state>
      0
      <j>0</j>
    </program_state>
  </step>
  <step>
    <rule>24</rule>
    <program_state>
      0
      <j>0</j>
    </program_state>
  </step>
  <step>
    <rule>25</rule>
    <program_state>
      0
      <j>0</j>
    </program_state>
  </step>
  <step>
    <rule>1</rule>
    <program_state>
      0
      <j>0</j>
    </program_state>
  </step>
  <step>
    <rule>12</rule>
    <program_state>
```

```
      0
      <j>0</j>
    </program_state>
  </step>
  <step>
    <rule>22</rule>
    <program_state>
      0
      <j>0</j>
    </program_state>
  </step>
  <step>
    <rule>26</rule>
    <program_state>
      0
      <j>0</j>
    </program_state>
  </step>
</answer>

## Notes:
- Each '<step>' must correspond to **exactly one K-semantics re-write
rule** that is needed to evaluate a statement in the given program.
- Only rules that have names indicated in '[]' adjacent to it must be
reported in the answer.
- The '<rule>' must indicate a rule used in the evaluation of a
statement.
- The '<program_state>' must represent the **entire program state
immediately after** the execution of that rule.
- The program state must list **all variables currently in scope**,
using the variable names as XML tags and their current values as tag
content.
- Include variables even if they did not change.
- Do not skip any step or merge multiple steps into one.
- Do not skip any rules (including those used to reduce expressions and
variables) that are needed to evaluate the program.

**Non-CoT**: Only output the '<answer>' XML block.  Do not include
explanations, comments, or any other text.
**CoT**: Explain your reasoning step-by-step **before** answering.  Wrap
your reasoning in '<reason>' tags.  Note that you **MUST** wrap your
reasoning steps with '<reason>' tags, the prediction with '<answer>'
tags.
```

## D.4 KEYWORDOBF OBFUSCATION TABLE

We provide the complete list of the mapping between the keywords and operators from standard
PL semantics to the Caucasian-Albanian symbols of KeywordObf in Table 12. The keywords
and operators in the original IMP program ($p$) under standard semantics will be replaced with the
corresponding Caucasian-Albanian symbols to get the semantically equivalent transformed program
($p'_{ko}$) under KeywordObf semantics.

**Table 12:** Complete list of obfuscations of operators and keywords in standard semantics to KeywordObf semantics.

| Type | Standard | $\rightarrow$ | KeywordObf |
|---|---|---|---|
| **Arithmetic** | + | | ꝺ |
| | − | | ꙡ |
| | * | | ꙅ |
| | / | | ꙇ |
| | % | | ꙃ |
| **Assignment** | = | | ꙍ |
| **Relational** | < | | ꙩ |
| | > | | ꜧ |
| | <= | | ꝑ |
| | >= | | ꝁ |
| | == | | ꝕ |
| | != | | ꝺ |
| **Logical** | ! | | ꙍ |
| | && | | ꝅ |
| | \|\| | | ʌ |
| **Keyword** | break | | ꙛ |
| | if-else | | � - ꝥ |
| | while | | ∠ |
| | halt | | ꝑ |
| | continue | | ꞇ |

# E    TASK EXTENDED ANALYSIS

## E.1    FINAL-STATE PREDICTION

This section analyzes (1) the impact of code-complexity metrics on LLM performance in the final-state prediction task, and (2) the average percentage of variables per program whose final states are predicted correctly.

### E.1.1    IMPACT OF CODE-COMPLEXITY METRICS

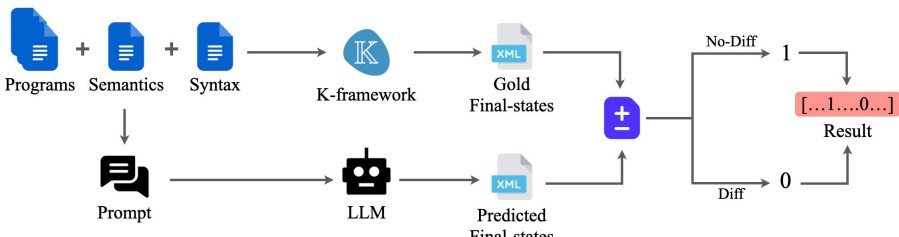

**(a)** Workflow of the final-state prediction task. IMP programs, along with optional semantics (K-framework or SOS) and syntax, are: (1) executed in the K-framework to obtain the gold final states of all declared variables, and (2) used to construct a prompt for the LLMs to predict those final states. The gold and predicted states are then compared, scored as 1 for a match and 0 otherwise, and accumulated into a result vector.

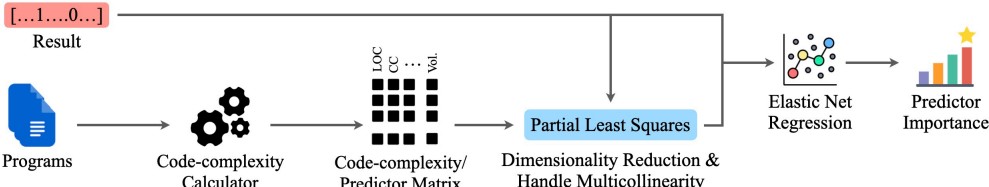

**(b)** Modeling LLM performance on IMP programs. We treat each LLM as a black box and apply **Elastic Net regression** using code-complexity metrics as predictors. **Partial Least Squares** (PLS) is employed for dimensionality reduction and to address multicollinearity. The magnitude and sign of the regression weights provide insight into the potential impact of each metric on the classifier's performance and hence to an extent the LLM's performance.

**Figure 7:** Analyzing the impact of different code-complexity metrics on LLM performance in the final-state prediction task.

Figure 7a illustrates the workflow of the final-state prediction task. An IMP program, together with optional semantics (K-framework or SOS) and syntax, is used both to construct prompts for the LLMs and to obtain gold final states by executing the program in the K-framework. The LLM's predicted final states are then compared with the gold states for each declared variable. A match is recorded as 1 (pass), and a mismatch as 0 (fail).

Different LLMs naturally excel on different IMP programs. To understand why an LLM may predict all final states correctly for one program but fail on another, we cast this task as a classification problem as shown in Figure 7b. Each IMP program is mapped to a predictor vector that characterizes its complexity, using the code-complexity metrics introduced earlier. Each predictor is then normalized using z-score normalization to ensure fair contribution from all the variables. The resulting predictor matrix, together with the LLM's binary result vector of passes and fails, is then used to train a classifier.

Because these complexity metrics are often highly correlated (multicollinearity), we apply Partial Least Squares (PLS) (Wold et al., 2001) for dimensionality reduction. Unlike the unsupervised Principal Component Analysis (PCA) (Wold et al., 1987), which identifies linear combinations of predictors that maximize variance, PLS is supervised: it reduces dimensionality by finding components that maximize the covariance between predictors and the response variables (the result

**Table 13:** Odds ratio per interquartile range ($\Theta(\Delta)$) for each code-complexity metric for the final-state prediction task **without semantics**. $\Theta(\Delta)$ for a metric is the odds ratio for a correct final-state prediction when that metric increases from its $25^{\text{th}}$ to its $75^{\text{th}}$ percentile, holding other metrics fixed. Reported only for models with <90% accuracy on the final-state prediction task (Table 5) to mitigate class imbalance. The largest absolute values in each row is shown in boldface font.

| Models | Control-flow | | | Data-flow | | Size | | | |
|---|---|---|---|---|---|---|---|---|---|
| | $\Omega_{\text{CC}}$ | $\hat{\Omega}_{\text{If}}$ | $\hat{\Omega}_{\text{Loop}}$ | $\Omega_{\text{DD}}$ | $\hat{\Omega}_{\text{Assign}}$ | $\Omega_{\text{Loc}}$ | $\Omega_{\text{Vol}}$ | $\Omega_{\text{Voc}}$ | $\hat{\Omega}_{\text{Trace}}$ |
| Human-Written | | | | | | | | | |
| Llama-3.3 70B | -19 | -5 | **-29** | -17 | -2 | -16 | -22 | -25 | -1 |
| Llama-3.3 70B-CoT | -21 | -14 | **-28** | -16 | -2 | -17 | -19 | -20 | -1 |
| Qwen2.5-Instruct 14B | -17 | -5 | **-27** | -16 | -2 | -14 | -20 | -25 | -1 |
| Qwen2.5-Instruct 14B-CoT | -25 | -18 | **-27** | -15 | -3 | -20 | -21 | -20 | -2 |
| Qwen2.5-Instruct 32B | -12 | -11 | -12 | -9 | -1 | -12 | -14 | **-17** | -1 |
| Qwen2.5-Instruct 32B-CoT | -23 | -7 | **-33** | -17 | -4 | -19 | -21 | -20 | -2 |
| GPT-4o-mini | -18 | -7 | **-30** | -16 | -2 | -13 | -18 | -22 | -1 |
| GPT-4o-mini-CoT | -15 | -2 | **-28** | -14 | -2 | -11 | -15 | -16 | -1 |
| DeepSeek-Qwen 14B | -13 | -10 | **-16** | -9 | -2 | -11 | -13 | -10 | -1 |
| DeepSeek-Llama 70B | -14 | -5 | **-22** | -12 | -3 | -11 | -14 | -10 | -2 |
| LLM-Translated | | | | | | | | | |
| QwQ 32B | -1 | -5 | 5 | **-20** | -4 | -13 | -20 | -7 | -4 |
| Fuzzer-Generated | | | | | | | | | |
| QwQ 32B | -25 | -25 | -25 | -14 | **-33** | -25 | -24 | -28 | -31 |
| GPT-5-mini | -21 | -14 | -19 | -12 | **-27** | -20 | -20 | -21 | -27 |
| Gemini-2.5-pro | -6 | -5 | -8 | -5 | **-12** | -6 | -6 | -5 | -12 |

vector). This makes PLS more suitable in our setting, as it better mitigates multicollinearity while preserving predictive power.

We next apply Elastic Net regression (Zou & Hastie, 2005) on the PLS-transformed predictors and the result vector to train a classifier. In regression, each predictor is assigned a coefficient whose magnitude reflects its relative importance and whose sign indicates whether it contributes positively or negatively to prediction accuracy. Elastic Net is chosen because it combines Lasso (Tibshirani, 1996) and Ridge (Hoerl & Kennard, 1970) regularization: the Lasso component drives irrelevant coefficients to zero, enabling feature selection, while the Ridge component shrinks correlated coefficients, thereby mitigating multicollinearity.

We now briefly describe the Elastic Net regression process to explain how we use the regression coefficients to determine the impact of different metrics. Let $n$, $p$, $\boldsymbol{y}$, and $\boldsymbol{X}$ be the total number of samples, the total number of predictors, the response vector, and the predictor matrix (we will use boldface font to denote vectors and matrices) respectively. Then,

$$\boldsymbol{y} \in \mathbb{R}^n, \quad y_i \in \{0, 1\}, \quad \boldsymbol{x_i} \in \mathbb{R}^p, \quad p_i(y_i = 1 | \boldsymbol{x_i}) = \frac{1}{1 + e^{-(\beta_0 + \boldsymbol{x_i}^\top \boldsymbol{\beta})}}$$

Where $p_i(y_i = 1 | \boldsymbol{x_i})$ along with $p_i(y_i = 0 | \boldsymbol{x_i}) = (1 - p_i(y_i = 1 | \boldsymbol{x_i}))$ represent the class-conditional probabilities and $\boldsymbol{\beta}$ is the vector of coefficients. The Elastic Net objective function for a Negative Log-Likelihood loss is given as (Friedman et al., 2010):

$$\arg\min_{\beta_0, \boldsymbol{\beta}} \left[ \frac{1}{n} \sum_{i=1}^{n} \left[ -y_i \log p_i - (1 - y_i) \log(1 - p_i) \right] + \lambda \underbrace{\sum_{j=1}^{p} \left[ \frac{1-\alpha}{2} \beta_j{}^2 + \alpha |\beta_j| \right]}_{\text{Ridge and Lasso penalties}} \right]$$

Let $\hat{\boldsymbol{\beta}}$ be the coefficient vector that minimizes this objective function. Then the percentage odds ratio (Agresti, 2013; Cornfield, 1951; Harrell, 2015) $\Theta$ for the inter-quartile-range $\Delta_j$ of the $j^{\text{th}}$ predictor can be computed as:

$$\Theta(\Delta_j) = 100 \times \left( \exp(\hat{\beta}_j \, \Delta_j) - 1 \right).$$

**Table 14:** Odds ratio per interquartile range ($\Theta(\Delta)$) for each code-complexity metric for the final-state prediction task **with the standard IMP semantics (K-framework and SOS)**. $\Theta(\Delta)$ for a metric is the odds ratio for a correct final-state prediction when that metric increases from its $25^{\text{th}}$ to its $75^{\text{th}}$ percentile, holding other metrics fixed. Reported only for models with <90% accuracy on the final-state prediction task (Table 5) to mitigate class imbalance. The largest absolute values in each row is shown in boldface font.

| Models | | Control-flow | | | Data-flow | | Size | | | |
|---|---|---|---|---|---|---|---|---|---|---|
| | | $\Omega_{\text{CC}}$ | $\hat{\Omega}_{\text{If}}$ | $\hat{\Omega}_{\text{Loop}}$ | $\Omega_{\text{DD}}$ | $\hat{\Omega}_{\text{Assign}}$ | $\Omega_{\text{Loc}}$ | $\Omega_{\text{Vol}}$ | $\Omega_{\text{Voc}}$ | $\hat{\Omega}_{\text{Trace}}$ |
| **Human-Written** | | | | | | | | | | |
| | LLAMA-3.3 70B | -25 | -4 | **-35** | -19 | -2 | -20 | -25 | -29 | -1 |
| | LLAMA-3.3 70B-CoT | -27 | -10 | **-33** | -20 | -3 | -24 | -26 | -25 | -2 |
| | QWEN2.5-INSTRUCT 14B | -24 | 0 | **-39** | -22 | -2 | -19 | -26 | -28 | -1 |
| | QWEN2.5-INSTRUCT 14B-CoT | -25 | -8 | **-35** | -16 | -3 | -20 | -22 | -22 | -2 |
| K | QWEN2.5-INSTRUCT 32B | -23 | -7 | **-35** | -19 | -2 | -19 | -25 | -30 | -1 |
| | QWEN2.5-INSTRUCT 32B-CoT | -21 | -15 | **-27** | -12 | -3 | -16 | -17 | -16 | -2 |
| | GPT-4O-MINI | -24 | -14 | **-32** | -21 | -2 | -22 | -27 | -30 | -1 |
| | GPT-4O-MINI-CoT | -21 | -14 | **-26** | -15 | -3 | -18 | -20 | -19 | -2 |
| | DEEPSEEK-QWEN 14B | **-29** | -21 | -27 | -20 | -3 | -26 | -27 | -23 | -2 |
| | DEEPSEEK-LLAMA 70B | -26 | -14 | **-33** | -14 | -5 | -19 | -19 | -15 | -3 |
| | LLAMA-3.3 70B | -24 | 0 | **-40** | -21 | -2 | -18 | -26 | -32 | -1 |
| | LLAMA-3.3 70B-CoT | -21 | -11 | **-32** | -16 | -4 | -18 | -20 | -21 | -2 |
| | QWEN2.5-INSTRUCT 14B | -19 | 2 | **-39** | -19 | -2 | -13 | -21 | -25 | -1 |
| | QWEN2.5-INSTRUCT 14B-CoT | **-26** | -17 | -25 | -16 | -3 | -22 | -22 | -20 | -2 |
| SOS | QWEN2.5-INSTRUCT 32B | -19 | -9 | **-28** | -16 | -1 | -17 | -21 | -27 | -1 |
| | QWEN2.5-INSTRUCT 32B-CoT | -19 | -14 | **-22** | -12 | -2 | -17 | -17 | -18 | -1 |
| | GPT-4O-MINI | -19 | 4 | **-37** | -19 | -2 | -16 | -23 | -29 | -1 |
| | GPT-4O-MINI-CoT | **-15** | -9 | -14 | -7 | -2 | -12 | -12 | -12 | -1 |
| | DEEPSEEK-QWEN 14B | -11 | -4 | **-14** | -9 | -1 | -10 | -12 | -9 | -1 |
| | DEEPSEEK-LLAMA 70B | -23 | -12 | **-32** | -14 | -5 | -18 | -21 | -18 | -3 |
| **LLM-Translated** | | | | | | | | | | |
| K | QwQ 32B | -11 | -6 | 7 | -22 | 0 | -24 | **-27** | -8 | 0 |
| SOS | QwQ 32B | -14 | -9 | -6 | **-28** | -4 | -18 | -20 | -1 | -4 |
| **Fuzzer-Generated** | | | | | | | | | | |
| | QwQ 32B | -21 | -27 | -25 | -10 | **-30** | -20 | -20 | -23 | -29 |
| K | GPT-5-MINI | -23 | -20 | -21 | -13 | **-31** | -22 | -22 | -23 | -30 |
| | GEMINI-2.5-PRO | -14 | -10 | -14 | -8 | **-21** | -13 | -13 | -14 | -21 |
| | QwQ 32B | -22 | -23 | -24 | -11 | **-31** | -22 | -21 | -25 | -30 |
| SOS | GPT-5-MINI | -22 | -19 | -21 | -11 | **-29** | -21 | -21 | -22 | -28 |
| | GEMINI-2.5-PRO | -7 | -20 | -24 | -3 | -25 | -7 | -7 | -7 | **-26** |

The percentage odds ratio per inter-quartile-range $\Theta(\Delta)$ gives the percentage change in the odds of the classifier's positive outcome (predicting a `1`) for the predictor ranging from its typical low value ($25^{\text{th}}$ percentile) to its typical high value ($75^{\text{th}}$ percentile) in the dataset when all other predictors are held constant. Thus if $\Theta(\Delta_j)$ for the $j^{\text{th}}$ predictor is -37%, this implies that one quartile increase in the $j^{\text{th}}$ predictor lowers the odds of the classifier's positive outcome by 37%.

To quantify each metric's effect on accuracy, we report the odds-ratio per interquartile range, $\Theta(\Delta)$, in Tables 13-14 for all LLMs with and without semantics (K-framework, SOS). Overall patterns are similar across settings. On the Human-Written dataset, $\Omega_{\text{Loop}}$ —the maximum executed loop-nesting depth—is the most influential predictor: larger $\Omega_{\text{Loop}}$ is associated with lower odds of a correct final-state prediction. On the LLM-Translated dataset, $\Omega_{\text{DD}}$ (data-flow complexity) and $\Omega_{\text{Vol}}$ (size) dominate without semantics; with semantics, $\Omega_{\text{DD}}$ remains dominant under SOS, whereas $\Omega_{\text{Vol}}$ dominates under K-framework. On the Fuzzer-Generated split, $\hat{\Omega}_{\text{Assign}}$ (total variable assignments) is the strongest predictor both without and with semantics, with one exception: for GEMINI-2.5-PRO under SOS, $\hat{\Omega}_{\text{Trace}}$ (execution-trace length) is most predictive. Collectively, these $\Theta(\Delta)$ trends suggest that increasing control-flow depth harms models on human code, whereas data-flow/size factors are more limiting on translated or synthetic code.

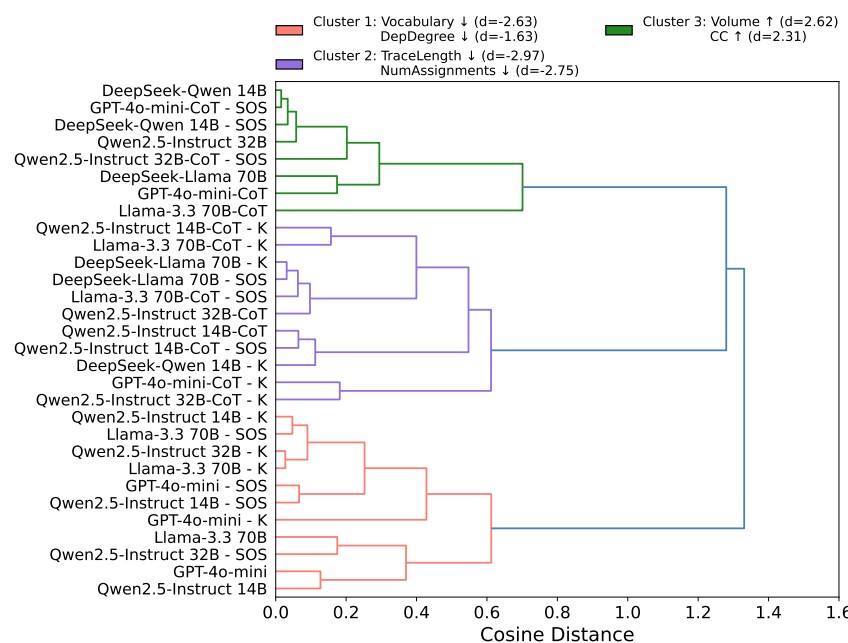

**Figure 8:** Dendrogram of models for the final-state prediction task on the Human-Written dataset under no-semantics and standard semantics (K-framework and SOS). We show the top two most distinguishable metrics per cluster, identified using the Cohen's d one-vs-rest test.

### E.1.2 COMPLEXITY-METRIC IMPACT PATTERNS

To identify if there is a pattern to how models perform on increasing different code-complexity metrics, we perform hierarchical clustering (Johnson, 1967) on the standardized regression coefficients ($\hat{\beta}_{\text{SD}}$) of the metrics for the models on the Human-Written dataset. We perform this for the no-semantics and with standard semantics (K-framework and SOS) cases. We use the cosine-distance as the pair-wise distance metric and the Cohen's d one-vs-rest test (Cohen, 1988) to identify the most distinguishing metric of each cluster. Figure 8 shows the dendrogram (Sokal & Rohlf, 1962) of the clustering process.

We see that there are three clusters. All the non-reasoning models without CoT prompting are in *Cluster 1* with the exception of QWEN2.5-INSTRUCT 32B (under no-semantics case). Cluster 1 responds more negatively to increases in the complexity metrics Vocabulary ($\Omega_{\text{Voc}}$) and DepDegree ($\Omega_{\text{DD}}$) relative to the other two clusters. *Cluster 2* contains only the reasoning models and the non-reasoning models with CoT prompting. It predominantly contains models under the K-framework semantics and responds more negatively to the dynamically computed metrics, TraceLength ($\hat{\Omega}_{\text{Trace}}$) and NumAssignments ($\hat{\Omega}_{\text{Assign}}$) relative to the rest of the clusters. The last cluster, *Cluster 3* also only contains reasoning models and non-reasoning models with CoT prompting (QWEN2.5-INSTRUCT 32B is an exception). It predominantly contains models under SOS semantics and responds positively to increases in the metrics, Volume ($\hat{\Omega}_{\text{Vol}}$) and cyclomatic-code complexity ($\Omega_{\text{CC}}$) relative to the rest.

### E.1.3 AVERAGE PERCENTAGE OF VARIABLES PREDICTED CORRECTLY

**Table 15:** Average percentage of variables predicted correctly per program on the final-state prediction task. Results are shown for both, SOS and K-semantics, under standard and nonstandard variants, across the Human-Written, LLM-Translated, and Fuzzer-Generated datasets. The best performing models in every column within a dataset are shown in boldface font. Only the top three best performing models (from different families) on the Human-Written dataset are evaluated on the LLM-Translated and the Fuzzer-Generated datasets.

| | Models | $p$ | K-semantics | | | SOS | | |
|---|---|---|---|---|---|---|---|---|
| | | | $(s,p)$ | $(s'_{ks}, p'_{ks})$ | $(s'_{ko}, p'_{ko})$ | $(s,p)$ | $(s'_{ks}, p'_{ks})$ | $(s'_{ko}, p'_{ko})$ |
| | **Human-Written** | | | | | | | |
| Non-reasoning | QWEN2.5-INSTRUCT 14B | 70 | 67 | 37 | 53 | 67 | 33 | 50 |
| | QWEN2.5-INSTRUCT 14B-CoT | 85 | 83 | 36 | 75 | 82 | 35 | 63 |
| | QWEN2.5-INSTRUCT 32B | 77 | 69 | 32 | 53 | 71 | 32 | 55 |
| | QWEN2.5-INSTRUCT 32B-CoT | 90 | 89 | 39 | 78 | 84 | 33 | 65 |
| | LLAMA-3.3 70B | 70 | 66 | 38 | 52 | 64 | 34 | 52 |
| | LLAMA-3.3 70B-CoT | 87 | 86 | 33 | 78 | 86 | 28 | 66 |
| | GPT-4O-MINI | 67 | 64 | 38 | 47 | 61 | 38 | 41 |
| | GPT-4O-MINI-CoT | 75 | 89 | 30 | 62 | 82 | 31 | 54 |
| Reasoning | DEEPSEEK-QWEN 14B | 66 | 83 | 27 | 53 | 60 | 20 | 43 |
| | DEEPSEEK-QWEN 32B | 85 | 97 | 45 | 85 | 98 | 36 | 88 |
| | DEEPSEEK-LLAMA 70B | 81 | 92 | 33 | 73 | 90 | 34 | 65 |
| | QwQ 32B | 94 | 99 | 82 | 91 | 100 | 38 | 92 |
| | o3-MINI | 95 | **100** | 59 | 92 | **100** | 74 | 98 |
| | GPT-5-MINI | **100** | 100 | 86 | **97** | 100 | 85 | 99 |
| | GEMINI-2.5-PRO | 93 | 100 | **98** | 97 | 100 | **99** | **100** |
| | **LLM-Translated** | | | | | | | |
| | QwQ 32B | 90 | 96 | 66 | 86 | 95 | 45 | 87 |
| | GPT-5-MINI | **98** | 98 | 88 | 96 | **98** | 81 | 97 |
| | GEMINI-2.5-PRO | 96 | **98** | 95 | 96 | 98 | 96 | 97 |
| | **Fuzzer-Generated** | | | | | | | |
| | QwQ 32B | 65 | 70 | 7 | 22 | 69 | 0 | 17 |
| | GPT-5-MINI | 91 | 82 | 22 | 33 | 84 | 33 | 34 |
| | GEMINI-2.5-PRO | **96** | **94** | **53** | **85** | **95** | **71** | **82** |

## E.2 SEMANTIC-RULE PREDICTION

In this section, we discuss: (1) how the statements sampled from IMP programs are processed for the PredRule task, (2) identify the most mis-predicted rule (first-point-of-mismatch) categories in the PredRule task, and (3) the rule categories the models are most likely to predict correctly.

### E.2.1 PROCESSING IMP STATEMENTS FOR PREDRULE

**Table 16:** Processing of statements sampled from IMP programs for the PredRule task.

| Type | Statement | State | PredRule Program | PredRule State |
|------|-----------|-------|------------------|----------------|
| Declaration | `int ;` | $\sigma$ | `int ;` | $\sigma$ |
| Assignment | ` = <EXP>;` | $\sigma$ | ` = <EXP>;` | $\sigma$ |
| While | `while(<PREDICATE>)`
`{`
`    <BODY>`
`};` | $\sigma$ | `while(<PREDICATE>)`
`{`
`  - <BODY>`
`  + halt;`
`};` | $\sigma$ |
| If-else | `if(<PREDICATE>)`
`{`
`    <BODY>`
`}`
`else`
`{`
`    <BODY>`
`};` | $\sigma$ | `if(<PREDICATE>)`
`{`
`  - <BODY>`
`  + halt;`
`}`
`else`
`{`
`  - <BODY>`
`  + halt;`
`};` | $\sigma$ |
| Halt | `halt;` | $\sigma$ | `halt;` | $\sigma$ |
| Break | `while(<PREDICATE>)`
`{`
`    ...`
`    break;`
`    ...`
`};` | $\sigma$ | `while(<PREDICATE>)`
`{`
`  - ...`
`    break;`
`    ...`
`};` | $\sigma$ |
| Continue | `while(<PREDICATE>)`
`{`
`    ...`
`    continue;`
`    ...`
`};` | $\sigma$ | `- while(<PREDICATE>)`
`+ while(<PREDICATE> && (ble != 1))`
`{`
`  - ...`
`  + ble = ble + 1;`
`    continue;`
`    ...`
`};` | $\sigma \cup \{\text{ble} : 0\}$ |

The objective of the PredRule task is to challenge LLMs with predicting the ordered sequence of semantic rules that is required to evaluate an IMP statement when the program state before the execution of that statement is given. Ideally, we want to avoid requiring the LLMs from needing to track program state since that capability is specifically tested for in the PredTrace task and we want to avoid any overlaps/redundancies. This is trivial for statements that are self-contained, such as `declaration`, `assignment`, and `halt`. However statements such as `while`, `if-else`, `break`, and `continue` require some processing to make them suitable for this task.

Table 16 shows how each type of statement is processed to make it suitable for the PredRule task. The primary objective behind processing is to make edits to the sampled statements such that they can be completely evaluated by requiring the least amount of program state updates. The first, second, and third columns lists the type of the sampled statement, its minimal representative skeleton, and the program state captured before its evaluation respectively. The fourth and the fifth columns list the sampled statement after processing and the corresponding processed program state which can now be used in the PredRule task. For the sampled `declaration`, `assignment`, and `halt` statements, the statements and the collected program state before their executions are used as is in the PredRule task because their evaluation does not require tracking program state nor do they require the execution of other statements. For `while` statements, we replace the body with a `halt` statement. This removes any possibility of needing state updates to correctly and completely evaluate the `while` statement. A similar approach is used for processing the `if-else` statement. For the `break` statement, we capture its closest enclosing loop and remove all statements from its body up until the `break` statement. A similar approach is taken for processing the `continue` statement but in addition, we modify the loop guard such that the loop executes for exactly one iteration which allows us to observe the semantic rules predicted by the models for evaluating the `continue` statement with requiring just one state update thereby ensuring minimal overlap with the PredTrace task.

Since the PredRule task is scoped to a statement level of granularity, it is relatively agnostic to the complexity of the program as a whole.

### E.2.2 Most Mis-Predicted Rules

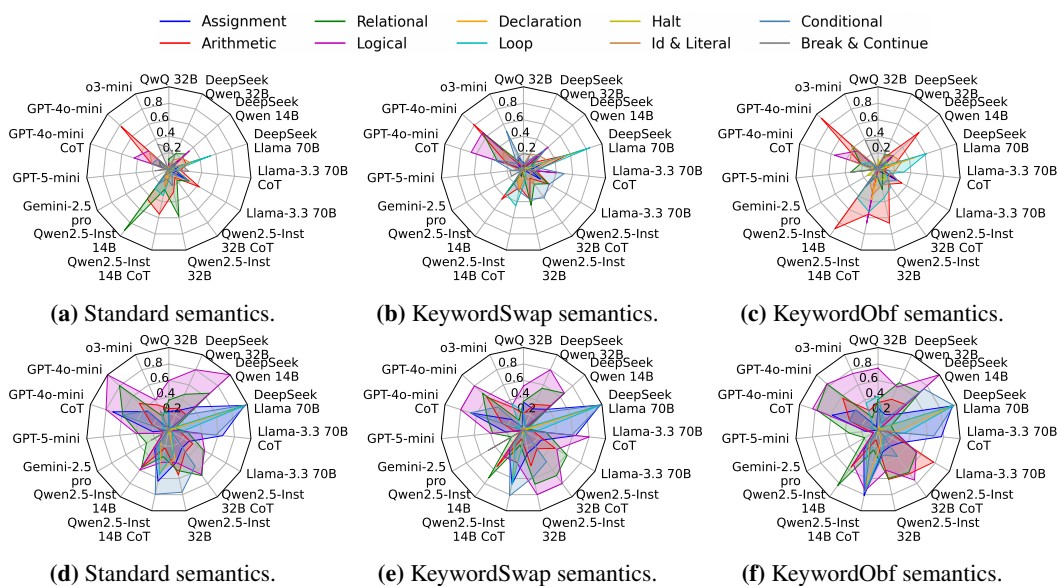

**(a)** Standard semantics.     **(b)** KeywordSwap semantics.     **(c)** KeywordObf semantics.

**(d)** Standard semantics.     **(e)** KeywordSwap semantics.     **(f)** KeywordObf semantics.

**Figure 9:** First-point-of-mismatch rate by category for the semantic-rule prediction task with the K-framework semantics (above) and SOS (below) on the Human-Written dataset.

To identify the semantic rules that models struggle with, we compute the *first-point-of-mismatch rate* for each rule, which is the frequency of the rule as the first mismatch between ground truth and the model prediction, relative to its total number of occurrences in the PredRule dataset. We group the rules into the following categories: *Assignment*, *Relational*, *Declaration*, *Halt*, *Conditional*, *Arithmetic*, *Logical*, *Loop*, *Id*, and *Break & Continue*. The mapping between the semantic rules and these categories for the K-framework and SOS is shown in Table 17. The first-point-of-mismatch rate for a category is the maximum across all the rules within this

**Table 17:** Rule categorization for PredRule task.

| Category | K-framework | SOS |
|---|---|---|
| Assignment | Rule 21 | Rules 4 - 6 |
| Arithmetic | Rules 3 - 11 | Rules 7 - 27 |
| Relational | Rules 12 - 17 | Rules 28 - 51 |
| Logical | Rules 18 - 20 | Rules 52 - 62 |
| Declaration | Rule 36 | Rule 3 |
| Loop | Rules 24 - 25 | Rules 67 - 70 & Rule 77 |
| Break & Continue | Rules 27 - 35 | Rules 71 - 76 |
| Halt | Rule 26 | Rule 78 |
| Id | Rules 1 - 2 | Rules 1 - 2 |
| Conditional | Rules 22 - 23 | Rules 64 - 66 |

ble 17. The first-point-of-mismatch rate for a category is the maximum across all the rules within this

category. Figure 9 shows the first-point-of-mismatch rate across categories for all the models on the Human-Written dataset for the standard and nonstandard semantics, for both their K-framework and SOS variants.

Firstly, we observe that models in general mis-predict rules to a larger extent for SOS relative to when provided with the K-framework semantics.

### E.2.3 MOST CORRECT RULES

To identify which rules the LLMs are most likely to predict correctly in the PredRule task, we compute the frequency with which each rule appears in the ground-truth rule list before the first-point-of-mismatch with the predicted list. These frequencies are then grouped by rule category, and the most-correct category rate is defined as the maximum frequency among its constituent rules. Figure 10 highlights the categories most accurately predicted by the LLMs. We see that most models are likely to predict rules belonging to `Halt` and `Declaration` categories correctly.

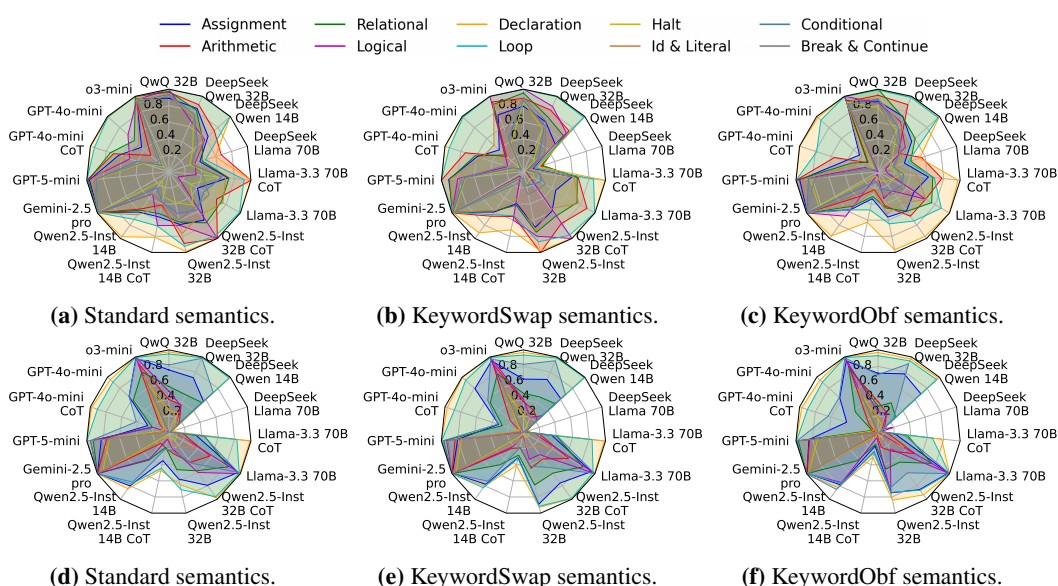

**(a)** Standard semantics.    **(b)** KeywordSwap semantics.    **(c)** KeywordObf semantics.

**(d)** Standard semantics.    **(e)** KeywordSwap semantics.    **(f)** KeywordObf semantics.

**Figure 10:** Most correctly predicted rules by category. All rules up until the first-point-of-mismatch between ground truth and predicted is defined as predicted correctly.

### E.3 EXECUTION-TRACE PREDICTION

In this section we perform extended analysis on the execution-trace prediction task through: (1) comparing only the final-states from the predicted and gold execution-traces, (2) identifying what percentages of the predicted and gold execution-traces match, and finally, (3) computing an approximate match metric between the predicted and gold execution-traces.

### E.3.1 FINAL-STATE ONLY COMPARISON

In the PredTrace task, we challenge the models with predicting the complete execution traces of the given programs. Some of the reasons the models can perform poorly on this task are due to: 1) predicting the program state incorrectly (computation error), 2) error in predicting the next statement to execute (control-flow error), 3) incorrect choice of semantic rules needed to evaluate a statement, and 4) failing to apply all the semantic rules necessary to evaluate a statement. To analyze this, we perform a final-state only comparison where we compare only the program states from the last step of the predicted and gold execution-traces. The results are shown in Table 18. We see that these numbers are significantly higher than the models' performances on the actual PredTrace task (Table 7). Therefore, computation related errors may not be the only type of errors models make on the PredTrace task Furthermore, we observe that most models' generally perform worse under

**Table 18:** Models' final-state-match accuracies on the PredTrace task with SOS and K-semantics.

| Models | IMP-K | | | IMP-SOS | | |
|---|---|---|---|---|---|---|
| | $\sigma(s,p)$ | $\sigma(s'_{ks},p'_{ks})$ | $\sigma(s'_{ko},p'_{ko})$ | $\sigma(s,p)$ | $\sigma(s'_{ks},p'_{ks})$ | $\sigma(s'_{ko},p'_{ko})$ |
| **Non-reasoning** | | | | | | |
| LLAMA-3.3 70B | 19 | 4 | 10 | 9 | 2 | 4 |
| LLAMA-3.3 70B-CoT | 19 | 3 | 14 | 14 | 2 | 12 |
| QWEN2.5-INSTRUCT 14B | 15 | 3 | 6 | 4 | 2 | 0 |
| QWEN2.5-INSTRUCT 14B-CoT | 22 | 2 | 12 | 9 | 2 | 3 |
| QWEN2.5-INSTRUCT 32B | 20 | 2 | 10 | 17 | 2 | 4 |
| QWEN2.5-INSTRUCT 32B-CoT | 27 | 3 | 15 | 27 | 1 | 8 |
| GPT-4O-MINI | 7 | 4 | 2 | 4 | 2 | 0 |
| GPT-4O-MINI-CoT | 21 | 3 | 10 | 16 | 2 | 1 |
| **Reasoning** | | | | | | |
| DEEPSEEK-QWEN 14B | 40 | 8 | 22 | 33 | 2 | 13 |
| DEEPSEEK-QWEN 32B | 47 | 26 | 31 | 36 | 3 | 25 |
| DEEPSEEK-LLAMA 70B | 10 | 1 | 7 | 1 | 0 | 2 |
| GEMINI-2.5-PRO | 85 | **78** | **81** | 77 | 73 | 73 |
| O3-MINI | 68 | 11 | 39 | 64 | 54 | 53 |
| QWQ 32B | 67 | 58 | 42 | 53 | 12 | 34 |
| GPT-5-MINI | **88** | 55 | 77 | **84** | **76** | **80** |

KeywordSwap relative to the standard and KeywordObf semantics, which is similar to the trend observed on the PredState task.

### E.3.2 TRACE PERCENTAGE COMPARISON

### E.3.3 APPROXIMATE MATCH OF THE EXECUTION-TRACE

**Table 19:** Percentage of execution-trace match for PredTrace for $(s,p)$, $(s'_{ks}, p'_{ks})$, and $(s'_{ko}, p'_{ko})$ with SOS.

| | Models | <10% | <20% | <30% | <40% | <50% | <60% | <70% | <80% | <90% | <100% |
|---|---|---|---|---|---|---|---|---|---|---|---|
| | **(s,p)** | | | | | | | | | | |
| Non-reasoning | LLAMA-3.3 70B | 14 | 5 | 3 | 1 | 1 | 1 | 1 | 1 | 0 | 0 |
| | LLAMA-3.3 70B-CoT | 14 | 7 | 3 | 1 | 0 | 0 | 0 | 0 | 0 | 0 |
| | QWEN2.5-INSTRUCT 14B | 12 | 8 | 4 | 1 | 0 | 0 | 0 | 0 | 0 | 0 |
| | QWEN2.5-INSTRUCT 14B-CoT | 17 | 6 | 3 | 1 | 1 | 0 | 0 | 0 | 0 | 0 |
| | QWEN2.5-INSTRUCT 32B | 28 | 17 | 7 | 2 | 1 | 0 | 0 | 0 | 0 | 0 |
| | QWEN2.5-INSTRUCT 32B-CoT | 27 | 14 | 4 | 1 | 0 | 0 | 0 | 0 | 0 | 0 |
| | GPT-4O-MINI | 9 | 4 | 1 | 1 | 0 | 0 | 0 | 0 | 0 | 0 |
| | GPT-4O-MINI-CoT | 12 | 9 | 3 | 1 | 0 | 0 | 0 | 0 | 0 | 0 |
| Reasoning | DEEPSEEK-QWEN 14B | 20 | 9 | 3 | 0 | 0 | 0 | 0 | 0 | 0 | 0 |
| | DEEPSEEK-QWEN 32B | 27 | 13 | 7 | 2 | 0 | 0 | 0 | 0 | 0 | 0 |
| | DEEPSEEK-LLAMA 70B | 1 | 1 | 1 | 0 | 0 | 0 | 0 | 0 | 0 | 0 |
| | GEMINI-2.5-PRO | 57 | 46 | 39 | 34 | 34 | 33 | 33 | 33 | 33 | 32 |
| | O3-MINI | 37 | 24 | 17 | 13 | 9 | 8 | 7 | 6 | 6 | 5 |
| | QwQ 32B | 28 | 16 | 7 | 4 | 1 | 0 | 0 | 0 | 0 | 0 |
| | GPT-5-MINI | 56 | 38 | 30 | 25 | 21 | 19 | 19 | 18 | 18 | 17 |
| | **(s'_{ks}, p'_{ks})** | | | | | | | | | | |
| Non-reasoning | LLAMA-3.3 70B | 16 | 4 | 3 | 1 | 1 | 1 | 1 | 1 | 1 | 0 |
| | LLAMA-3.3 70B-CoT | 24 | 10 | 5 | 2 | 1 | 0 | 0 | 0 | 0 | 0 |
| | QWEN2.5-INSTRUCT 14B | 15 | 9 | 3 | 1 | 0 | 0 | 0 | 0 | 0 | 0 |
| | QWEN2.5-INSTRUCT 14B-CoT | 12 | 4 | 0 | 0 | 0 | 0 | 0 | 0 | 0 | 0 |
| | QWEN2.5-INSTRUCT 32B | 30 | 15 | 7 | 3 | 0 | 0 | 0 | 0 | 0 | 0 |
| | QWEN2.5-INSTRUCT 32B-CoT | 31 | 15 | 6 | 1 | 0 | 0 | 0 | 0 | 0 | 0 |
| | GPT-4O-MINI | 7 | 4 | 1 | 1 | 0 | 0 | 0 | 0 | 0 | 0 |
| | GPT-4O-MINI-CoT | 8 | 3 | 2 | 1 | 0 | 0 | 0 | 0 | 0 | 0 |
| Reasoning | DEEPSEEK-QWEN 14B | 21 | 9 | 3 | 1 | 0 | 0 | 0 | 0 | 0 | 0 |
| | DEEPSEEK-QWEN 32B | 31 | 16 | 7 | 1 | 0 | 0 | 0 | 0 | 0 | 0 |
| | DEEPSEEK-LLAMA 70B | 5 | 2 | 2 | 1 | 0 | 0 | 0 | 0 | 0 | 0 |
| | GEMINI-2.5-PRO | 58 | 48 | 41 | 37 | 36 | 36 | 36 | 36 | 35 | 35 |
| | O3-MINI | 35 | 20 | 14 | 10 | 7 | 5 | 5 | 4 | 4 | 3 |
| | QwQ 32B | 21 | 13 | 6 | 2 | 1 | 0 | 0 | 0 | 0 | 0 |
| | GPT-5-MINI | 55 | 38 | 30 | 23 | 21 | 17 | 17 | 16 | 15 | 15 |
| | **(s'_{ko}, p'_{ko})** | | | | | | | | | | |
| Non-reasoning | LLAMA-3.3 70B | 14 | 8 | 1 | 1 | 0 | 0 | 0 | 0 | 0 | 0 |
| | LLAMA-3.3 70B-CoT | 12 | 7 | 3 | 1 | 1 | 1 | 0 | 0 | 0 | 0 |
| | QWEN2.5-INSTRUCT 14B | 8 | 6 | 2 | 1 | 0 | 0 | 0 | 0 | 0 | 0 |
| | QWEN2.5-INSTRUCT 14B-CoT | 12 | 6 | 1 | 0 | 0 | 0 | 0 | 0 | 0 | 0 |
| | QWEN2.5-INSTRUCT 32B | 22 | 10 | 6 | 3 | 0 | 0 | 0 | 0 | 0 | 0 |
| | QWEN2.5-INSTRUCT 32B-CoT | 27 | 14 | 6 | 1 | 0 | 0 | 0 | 0 | 0 | 0 |
| | GPT-4O-MINI | 8 | 5 | 2 | 1 | 0 | 0 | 0 | 0 | 0 | 0 |
| | GPT-4O-MINI-CoT | 9 | 6 | 2 | 2 | 0 | 0 | 0 | 0 | 0 | 0 |
| Reasoning | DEEPSEEK-QWEN 14B | 15 | 8 | 3 | 1 | 0 | 0 | 0 | 0 | 0 | 0 |
| | DEEPSEEK-QWEN 32B | 26 | 13 | 7 | 3 | 2 | 1 | 1 | 1 | 1 | 1 |
| | DEEPSEEK-LLAMA 70B | 3 | 1 | 1 | 1 | 0 | 0 | 0 | 0 | 0 | 0 |
| | GEMINI-2.5-PRO | 56 | 45 | 39 | 36 | 35 | 35 | 35 | 35 | 35 | 35 |
| | O3-MINI | 32 | 18 | 11 | 6 | 4 | 2 | 2 | 2 | 2 | 2 |
| | QwQ 32B | 22 | 13 | 6 | 2 | 1 | 0 | 0 | 0 | 0 | 0 |
| | GPT-5-MINI | 53 | 39 | 30 | 25 | 22 | 20 | 20 | 18 | 17 | 17 |

**Table 20:** Percentage of execution-trace match for PredTrace for $(s,p)$, $(s'_{ks}, p'_{ks})$, and $(s'_{ko}, p'_{ko})$ with the K-framework semantics.

| | Models | <10% | <20% | <30% | <40% | <50% | <60% | <70% | <80% | <90% | <100% |
|---|---|---|---|---|---|---|---|---|---|---|---|
| | | | | | $(s,p)$ | | | | | | |
| Non-reasoning | LLAMA-3.3 70B | 28 | 21 | 14 | 10 | 9 | 6 | 4 | 3 | 2 | 2 |
| | LLAMA-3.3 70B-CoT | 18 | 15 | 11 | 9 | 9 | 7 | 7 | 6 | 6 | 6 |
| | QWEN2.5-INSTRUCT 14B | 14 | 12 | 10 | 7 | 3 | 0 | 0 | 0 | 0 | 0 |
| | QWEN2.5-INSTRUCT 14B-CoT | 20 | 15 | 12 | 8 | 4 | 1 | 0 | 0 | 0 | 0 |
| | QWEN2.5-INSTRUCT 32B | 37 | 22 | 15 | 8 | 4 | 1 | 0 | 0 | 0 | 0 |
| | QWEN2.5-INSTRUCT 32B-CoT | 35 | 22 | 15 | 7 | 4 | 2 | 2 | 1 | 1 | 1 |
| | GPT-4O-MINI | 35 | 23 | 15 | 10 | 6 | 2 | 1 | 0 | 0 | 0 |
| | GPT-4O-MINI-CoT | 34 | 21 | 14 | 8 | 4 | 2 | 0 | 0 | 0 | 0 |
| Reasoning | DEEPSEEK-QWEN 14B | 30 | 19 | 12 | 7 | 3 | 1 | 1 | 1 | 1 | 1 |
| | DEEPSEEK-QWEN 32B | 38 | 26 | 20 | 15 | 12 | 10 | 9 | 8 | 8 | 8 |
| | DEEPSEEK-LLAMA 70B | 9 | 8 | 7 | 7 | 6 | 4 | 4 | 4 | 3 | 3 |
| | GEMINI-2.5-PRO | 52 | 35 | 31 | 26 | 26 | 25 | 25 | 25 | 25 | 25 |
| | O3-MINI | 45 | 29 | 25 | 21 | 21 | 20 | 20 | 19 | 19 | 19 |
| | QwQ 32B | 43 | 31 | 27 | 23 | 21 | 20 | 20 | 19 | 18 | 18 |
| | GPT-5-MINI | 50 | 33 | 28 | 25 | 23 | 22 | 21 | 21 | 20 | 20 |
| | | | | | $(s'_{ks}, p'_{ks})$ | | | | | | |
| Non-reasoning | LLAMA-3.3 70B | 31 | 17 | 9 | 7 | 4 | 3 | 2 | 1 | 0 | 0 |
| | LLAMA-3.3 70B-CoT | 31 | 19 | 14 | 10 | 7 | 4 | 2 | 0 | 0 | 0 |
| | QWEN2.5-INSTRUCT 14B | 7 | 7 | 6 | 4 | 2 | 0 | 0 | 0 | 0 | 0 |
| | QWEN2.5-INSTRUCT 14B-CoT | 22 | 17 | 12 | 9 | 4 | 2 | 1 | 0 | 0 | 0 |
| | QWEN2.5-INSTRUCT 32B | 38 | 21 | 16 | 9 | 4 | 2 | 1 | 0 | 0 | 0 |
| | QWEN2.5-INSTRUCT 32B-CoT | 36 | 22 | 14 | 6 | 4 | 1 | 1 | 1 | 1 | 1 |
| | GPT-4O-MINI | 35 | 23 | 14 | 8 | 5 | 2 | 1 | 1 | 0 | 0 |
| | GPT-4O-MINI-CoT | 28 | 16 | 11 | 6 | 3 | 1 | 1 | 0 | 0 | 0 |
| Reasoning | DEEPSEEK-QWEN 14B | 35 | 20 | 13 | 7 | 3 | 0 | 0 | 0 | 0 | 0 |
| | DEEPSEEK-QWEN 32B | 34 | 24 | 17 | 10 | 7 | 4 | 3 | 2 | 2 | 2 |
| | DEEPSEEK-LLAMA 70B | 4 | 4 | 4 | 2 | 2 | 1 | 1 | 1 | 0 | 0 |
| | GEMINI-2.5-PRO | 53 | 35 | 31 | 26 | 26 | 25 | 25 | 25 | 25 | 25 |
| | O3-MINI | 45 | 27 | 21 | 16 | 14 | 9 | 7 | 4 | 3 | 3 |
| | QwQ 32B | 35 | 27 | 23 | 20 | 19 | 17 | 17 | 17 | 16 | 16 |
| | GPT-5-MINI | 47 | 31 | 27 | 23 | 21 | 18 | 17 | 15 | 14 | 14 |
| | | | | | $(s'_{ko}, p'_{ko})$ | | | | | | |
| Non-reasoning | LLAMA-3.3 70B | 27 | 19 | 15 | 12 | 10 | 6 | 5 | 4 | 4 | 3 |
| | LLAMA-3.3 70B-CoT | 17 | 16 | 13 | 10 | 6 | 4 | 3 | 3 | 3 | 3 |
| | QWEN2.5-INSTRUCT 14B | 11 | 9 | 7 | 6 | 3 | 0 | 0 | 0 | 0 | 0 |
| | QWEN2.5-INSTRUCT 14B-CoT | 18 | 13 | 8 | 6 | 2 | 0 | 0 | 0 | 0 | 0 |
| | QWEN2.5-INSTRUCT 32B | 32 | 19 | 13 | 7 | 3 | 0 | 0 | 0 | 0 | 0 |
| | QWEN2.5-INSTRUCT 32B-CoT | 29 | 16 | 11 | 7 | 3 | 0 | 0 | 0 | 0 | 0 |
| | GPT-4O-MINI | 31 | 19 | 11 | 8 | 4 | 2 | 1 | 1 | 0 | 0 |
| | GPT-4O-MINI-CoT | 23 | 15 | 10 | 6 | 3 | 0 | 0 | 0 | 0 | 0 |
| Reasoning | DEEPSEEK-QWEN 14B | 29 | 17 | 12 | 6 | 3 | 0 | 0 | 0 | 0 | 0 |
| | DEEPSEEK-QWEN 32B | 35 | 23 | 17 | 10 | 7 | 5 | 5 | 4 | 3 | 3 |
| | DEEPSEEK-LLAMA 70B | 9 | 7 | 7 | 6 | 5 | 4 | 3 | 3 | 3 | 3 |
| | GEMINI-2.5-PRO | 52 | 35 | 32 | 28 | 26 | 26 | 26 | 26 | 26 | 25 |
| | O3-MINI | 43 | 26 | 23 | 19 | 17 | 16 | 15 | 14 | 14 | 13 |
| | QwQ 32B | 42 | 29 | 25 | 22 | 21 | 20 | 19 | 17 | 16 | 15 |
| | GPT-5-MINI | 48 | 33 | 29 | 25 | 24 | 22 | 22 | 21 | 20 | 17 |

**Table 21:** Models' approximate-match accuracies on the PredTrace task with SOS and K-semantics.

| | Models | IMP-K | | | IMP-SOS | | |
|---|---|---|---|---|---|---|---|
| | | $\sim(s,p)$ | $\sim(s'_{ks}, p'_{ks})$ | $\sim(s'_{ko}, p'_{ko})$ | $\sim(s,p)$ | $\sim(s'_{ks}, p'_{ks})$ | $\sim(s'_{ko}, p'_{ko})$ |
| Non-reasoning | LLAMA-3.3 70B | 4 | 0 | 4 | 0 | 0 | 0 |
| | LLAMA-3.3 70B-CoT | 8 | 0 | 7 | 2 | 0 | 3 |
| | QWEN2.5-INSTRUCT 14B | 1 | 0 | 0 | 0 | 1 | 0 |
| | QWEN2.5-INSTRUCT 14B-CoT | 1 | 0 | 1 | 1 | 0 | 0 |
| | QWEN2.5-INSTRUCT 32B | 3 | 1 | 3 | 1 | 0 | 1 |
| | QWEN2.5-INSTRUCT 32B-CoT | 7 | 1 | 5 | 1 | 0 | 1 |
| | GPT-4O-MINI | 1 | 1 | 0 | 0 | 0 | 0 |
| | GPT-4O-MINI-CoT | 5 | 0 | 3 | 1 | 0 | 0 |
| Reasoning | DEEPSEEK-QWEN 14B | 12 | 5 | 7 | 3 | 0 | 2 |
| | DEEPSEEK-QWEN 32B | 17 | 11 | 13 | 10 | 1 | 8 |
| | DEEPSEEK-LLAMA 70B | 6 | 1 | 4 | 1 | 0 | 1 |
| | GEMINI-2.5-PRO | **25** | **25** | **26** | **66** | **60** | **66** |
| | O3-MINI | 21 | 5 | 16 | 27 | 23 | 22 |
| | QwQ 32B | 21 | 19 | 19 | 15 | 6 | 11 |
| | GPT-5-MINI | 21 | 15 | 19 | 46 | 43 | 42 |

## F    LIMITATIONS

PLSEMANTICSBENCH evaluates models on programs written in IMP which is limited in complex semantic features. Strong performance on PLSEMANTICSBENCH does not necessarily generalize to more sophisticated programming languages. Nonetheless, PLSEMANTICSBENCH serves as an initial step toward evaluating the effectiveness of using LLMs as interpreters.

We formalize the semantics of the IMP language using small-step Structural Operational Semantics (SOS) and K-semantics, which may not be the most suitable PL semantics representation for LLMs. Our choice of using formal semantics was to avoid ambiguities while describing the semantics as it is easier to check for in formal semantics. Investigating LLMs' performance under alternative semantics specification including natural language, big-step or denotational semantics, is left as future work.

## G    USE OF EXTERNAL ASSETS

In this work, we make use of several external assets, including datasets, and pretrained models. We acknowledge and credit the original creators of these assets as follows:

### G.1    DATA

We construct the Human-Written dataset by rewriting the existing code solutions from the following sources:

1. HumanEval-X
   (a) License: Apache 2.0
   (b) URL: `https://huggingface.co/datasets/THUDM/humaneval-x`
2. BabelCode MBPP
   (a) License: CC 4.0
   (b) URL: `https://huggingface.co/datasets/gabeorlanski/bc-mbpp`
3. CodeContests
   (a) License: CC 4.0
   (b) URL: `https://github.com/google-deepmind/code_contests`
4. Leetcode
   (a) We scrape only the ground-truth solutions and public test cases from leetcode. We use the collected problems for academic purposes only.
   (b) URL: `https://leetcode.com/`

We construct the LLM-Translated dataset by using QWEN2.5-INSTRUCT 32B to translate the C++ solutions to problems from:

1. CodeForces
   (a) License: CC 4.0
   (b) URL: `https://huggingface.co/datasets/open-r1/codeforces`

### G.2    MODELS

We evaluate LLMs designed for coding tasks and enhanced reasoning ability on our PLSEMANTICS-BENCH:

1. LLAMA-3.3 70B (Grattafiori et al., 2024),
   (a) License: llama3.3
   (b) URL:
       `https://huggingface.co/meta-llama/Llama-3.3-70B-Instruct`
2. Qwen2.5-Coder Models (Hui et al., 2024),

    (a) License: Apache 2.0

    (b) URLs:
```
https://huggingface.co/Qwen/Qwen2.5-Coder-32B-Instruct
https://huggingface.co/Qwen/Qwen2.5-Coder-14B-Instruct
```

3. DeepSeek-R1 distilled models (Guo et al., 2025)

    (a) License: MIT

    (b) URLs:
```
https://huggingface.co/deepseek-ai/DeepSeek-R1-Distill-Llama-70B
https://huggingface.co/deepseek-ai/DeepSeek-R1-Distill-Qwen-32B
https://huggingface.co/deepseek-ai/DeepSeek-R1-Distill-Qwen-14B
```

4. QwQ 32B (Team, 2025b)

    (a) License: Apache 2.0

    (b) URL: `https://huggingface.co/Qwen/QwQ-32B`

5. GEMINI-2.5-PRO. In this study, we utilized the GEMINI-2.5-PRO model provided by Google AI. The use of this model is subject to the Generative AI Preview Terms and Conditions, as outlined in the Google Cloud Service Specific Terms for Pre-GA Offerings.

    (a) URL: `https://cloud.google.com/terms/service-terms`

6. OpenAI Models. In this study, the use of OpenAI's models is subject to the term of use.

    (a) URL: `https://openai.com/policies/row-terms-of-use/`

