# OpenReview forum: "PLSEMANTICSBENCH: LARGE LANGUAGE MODELS AS PROGRAMMING LANGUAGE INTERPRETERS"
_ICLR.cc/2026/Conference — Submitted to ICLR 2026_

### Official Review · Reviewer_Cx12 · 2025-10-19

**Soundness:** 2
**Presentation:** 2
**Contribution:** 2
**Rating:** 2
**Confidence:** 5

**Summary:**

This paper introduces PLSEMANTICSBENCH, a benchmark to evaluate if LLMs can function as program interpreters based on given formal semantics. The authors evaluate various LLMs on programs written in the IMP language using two formalisms, Small-step Operational Semantics and K-semantics, across three tasks: predicting a program's final state, the sequence of semantic rules used, and the full execution trace, including both used rules and program states at each step. To differentiate true semantic understanding from memorization, they also introduce two "nonstandard" semantics, KeywordSwap and KeywordObf,  that alter the language's common rules.

The experiments mainly show that while LLMs perform well on coarse-grained tasks (final state prediction) with standard semantics, their performance drops significantly on fine-grained tasks and with nonstandard, altered semantics. This suggests that current LLMs have a shallow understanding of programming language semantics and rely heavily on pre-trained patterns rather than robustly interpreting the provided rules.

**Strengths:**

- **Experimental Design:** While there are previous efforts on evaluating the performance of execution reasoning of LLMs, this paper may be the first one that starts with formal semantics and asks the model to simulate the execution given probably unseen formal semantics. This design, together with the obfuscation techniques proposed, stress-test whether LLMs can perform true reasoning on the fly, instead of relying on remembering pre-trained knowledge.
- **Evaluation:** This paper evaluates various popular LLMs on different kinds of data to present comprehensive empirical results in their settings. Results highlight that LLMs cannot perform execution simulation on nonstandard semantics and finer-grained levels.

**Weaknesses:**

- **Lack of depth on the ML side:** While this paper empirically presents the limitations of current LLMs on execution simulation using non-standard formal semantics, it does not provide deeper insights into such performance issues to propose constructive future steps that can improve the performance. The major concern of using non-standard semantics is the pre-training bias, which is also pointed out in this paper. But the authors fail to further break down this issue to provide an in-depth analysis. For example, is this a bias that can be easily addressed by data-driven approaches? To me, the nearly perfect performance of Gemini-2.5-pro implies that this can work, because if the model can handle the normal "+" pretty well, then it must also handle the obfuscated versions very well if they are present in the training data. Then, this will no longer be an issue because we do not have many new programming languages coming out every day to analyze, and it is fully acceptable that LLMs can only simulate a limited set of pre-existing languages that they have been trained on. In summary, we should distinguish whether the poor performance comes from data issues or fundamentally expressiveness issues (i.e., whether the LLMs are fundamentally limited in simulating execution even with sufficient training efforts). I don't think it is the latter one as [previous works](https://arxiv.org/abs/2310.07923) point out that Transformers can simulate a Turing machine given sufficient decoding steps. Then, for the data issue, we have easy solutions, such as providing some few-shot examples for sufficient in-context learning, or doing SFT.
- **Unexplained Experimental Results:**
  - The paper finds that "providing semantics hurts the performance of even the reasoning models", but there is no further analysis of this phenomenon.
  - The paper shows that the performance on fine-grained tasks like step-wise prediction is much worse than the final-state prediction. Why does this happen? For human-written programs, Gemini-2.5-Pro does a great job on final-state prediction, but is very poor on step-wise prediction. Without further analysis, one may wonder if this result is reliable. For example, maybe the model fails to always produce its predictions in the correct XML format, or the evaluation based on exact match is not robust to slight variations and cannot reflect the model's true capability of step-wise reasoning.
- **Practical Significance:** This paper does not provide any practical guidance on how to improve the model's execution reasoning capability. It also does not connect the task of execution simulation to practical real-world tasks and explain how such a simulation given formal semantics can help with downstream tasks like debugging.

**Questions:**

Please see the weaknesses section above, where the questions and concerns are stated.

---

### Official Review · Reviewer_Zhzx · 2025-10-31

**Soundness:** 3
**Presentation:** 3
**Contribution:** 2
**Rating:** 4
**Confidence:** 3

**Summary:**

The authors ask whether LLMs can act as interpreters for programming languages by executing code based solely on formal semantics. They introduce a benchmark for this: a small imperative language (IMP) formalised via operational semantics and K-semantics. They define three tasks of increasing granularity:
* Predict final program state (PredState)
* Predict sequence of semantic rules applied (PredRule)
* Predict full execution trace (PredTrace)
They evaluate multiple strong LLMs across programs with standard semantics and deliberately mutated “non-standard” semantics.

**Key Contributions**
1. A new benchmark (PLSemanticsBench) for evaluating LLMs as programming language interpreters.
2. Empirical findings: LLMs do reasonably well on simple/familiar semantics, but performance drops sharply under non-standard semantics or full trace tasks.
3. Insights: The gap between code-generation ability and actual semantic understanding is large; providing formal semantics helps in some settings but not reliably in more complex ones.

**Why it matters**
This work moves beyond just "can the model write or complete code?" to "can the model *understand* what a program means and how it executes?" It surfaces deeper limitations in LLMs’ reasoning about code semantics.

**Strengths:**

* The paper introduces the idea of treating large language models (LLMs) as programming-language interpreters, i.e., executing programs purely by following formal semantics. Prior work on LLMs + code largely focused on code generation, summarization, or completion (e.g., generating correct outputs or translating code) rather than executing via formal semantics. By contrast, PLSemanticsBench uses operational semantics (small-step SOS and K-semantics) for a small imperative language (IMP). Benchmarks like EquiBench [A] focused on equivalence checking of programs (i.e., do two programs have the same behaviour) rather than requiring step-by-step semantic rule application. Thus the strength is that PLSemanticsBench pushes deeper: not just “can the model generate correct code” but “can the model understand and apply semantics”.
* The paper defines three tasks of increasing granularity: PredState (final state prediction), PredRule (predict which semantic rules applied), and PredTrace (complete execution trace). Many earlier code LLM evaluations stop at output / final-state correctness (e.g., code generation benchmarks). PLSemanticsBench explicitly tests the steps of execution (rule trace) which is more fine-grained. This is a strength because it opens up visibility into how much understanding the model has, beyond just what it predicts.
* The authors deliberately mutate semantics (KeywordSwap, KeywordObf) to see if models rely on memorisation of known semantics or genuinely apply semantics. Earlier work rarely injects semantic changes to test generalisation. For example EquiBench tests equivalence but under the same semantics; standard code-generation tasks assume fixed semantics. This robustness check is a clear strength: it helps isolate "memorised semantics vs flexible reasoning".
* The paper shows that models that perform well on standard semantics still suffer large drops under non-standard semantics, and that providing formal semantics helps for simple programs but can hurt for complex ones. Prior benchmarks (e.g., code generation or equivalence checking) may show model successes but less often expose where semantics understanding breaks. PLSemanticsBench surfaces these failure modes explicitly.

```
[A] EquiBench: Benchmarking Large Language Models' Reasoning about Program Semantics via Equivalence Checking, ArXiv 2025
```

**Weaknesses:**

* The benchmark uses a small imperative language IMP (a standard toy subset) which, although clearly defined via operational semantics (SOS and K-semantics) for the evaluation tasks, is still far removed from the complexity of real-world programming languages.
* The paper shows that performance drops under nonstandard semantics, indicating models rely on known semantics; not pure semantic reasoning. But it is unclear to what degree the models might still be leveraging memorised patterns of code execution rather than symbolically "interpreting" semantics. The benchmark stops short of fully disentangling these modes of failure (i.e., memorisation vs generalisation).
* The fact that providing the formal semantics sometimes hurts performance on more complex programs (as reported) suggests the evaluation might be conflating two orthogonal issues: semantics understanding and prompt/model capacity/length issues.
* While the three tasks (PredState, PredRule, PredTrace) provide increasing granularity, the PredRule and PredTrace tasks are very challenging and perhaps somewhat artificial in real-world interpreter use. It is unclear whether the ability to generate exact semantic rule sequences is necessary or aligned with practical interpreter use-cases (e.g., executing a program correctly).
* There is limited discussion of how prompt design, context length, tokenisation of semantics, or model input size affect results. Since semantics inclusion can cause prompt blow-up, it is possible that degraded performance on complex programs is due in part to prompt length/engine limitations rather than semantics comprehension per se.
* The paper might benefit from linking to more general work on program‐aided language models e.g. PAL [A] which considered coupling LLMs with symbolic interpreters. The connection of "LLM as interpreter" to "LLM + interpreter external + symbolic system" could be richer.

```
[A] PAL: Program-aided Language Models, ICML 2023
```

**Questions:**

* Is it possible to extend the benchmark to include one or more more realistic languages (for example, a subset of Java, Python, or C with features like dynamic memory, objects, exceptions) and the formal specification of key semantics (or a well-documented operational semantics fragment)?
* Is it possible to introduce tasks that involve richer semantics: e.g., aliasing, heap mutations, recursion, concurrency, exceptions? This would stress test model understanding beyond a toy imperative language.
* Is it possible to provide more justification for inclusion of the full trace task? In what downstream applications is full semantic trace generation valuable (web debugging, program verification, symbolic execution)? This will strengthen the alignment of the benchmark with real needs.
* Conduct controlled experiments varying prompt length and semantic specification length to isolate the impact of input size on model performance (i.e., is the drop in complex programs due to "too much input" rather than lack of semantics understanding?).
* Is it possible to provide further complexity‐ablations: for example, vary loop depth, data‐flow complexity, number of variables, branching factor systematically and report how each dimension affects performance? This enables a more nuanced understanding of "where performance breaks".
* Is it possible to provide guidelines for how to adapt the benchmark results to practical tool development: e.g., what levels of PredState performance might be sufficient for typical interpreter use, how trace correctness may be weighted vs final state, and what error handling mechanisms might be needed?

---

### Official Review · Reviewer_621n · 2025-11-03

**Soundness:** 3
**Presentation:** 3
**Contribution:** 2
**Rating:** 6
**Confidence:** 4

**Summary:**

This paper investigates whether Large Language Models (LLMs) can function as programming language interpreters, executing code based on formal semantics rather than just relying on pattern matching from their training data.

PLSemanticsBench uses the imperative language IMP (a subset of C) and formalizes its behavior using two standard semantic styles: small-step operational semantics (SOS) and rewriting-based K-semantics. The evaluation is conducted across 3 distinct datasets (Human-Written, LLM-Translated, and Fuzzer-Generated) and 3 tasks of increasing difficulty: predicting a program's final state (PredState), predicting the sequence of semantic rules applied (PredRule), and predicting the full execution trace (PredTrace). A key and novel element of their methodology is the introduction of nonstandard semantics, created by systematically mutating the standard rules (e.g., swapping the meaning of + and -). This is designed to test whether models are truly reasoning about the provided semantics or just recalling patterns of standard code execution.

The results show that while modern LLMs perform well on coarse-grained tasks under standard semantics, their performance degrades significantly when faced with nonstandard semantics. This suggests that their success is often driven by pre-trained knowledge rather than a robust, on-the-fly understanding of the formal rules.

**Strengths:**

## Novelty

**1. Rigorous Evaluation Framework:** The methodology of testing models against formal semantics, particularly with the inclusion of nonstandard semantics, is a novel approach to disentangling true semantic understanding from memorization. This provides a much-needed, more rigorous evaluation paradigm than existing benchmarks that often focus on surface-level code generation or execution prediction tasks.

**2. Introduction of Non-standard Semantics:** The use of semantic mutations (KeywordSwap and KeywordObf) is a particularly innovative and effective method for testing the robustness of LLM reasoning. By creating a scenario where models cannot rely on their pre-trained knowledge, the authors force a true test of on-the-fly semantic interpretation. This is a powerful technique that could be adopted by other researchers in the field to create more challenging and insightful evaluations.

## Soundness

**1. Comprehensive Experimental Design:** The experimental design is a major strength of this paper and deserves special recognition. The authors have created a remarkably comprehensive evaluation framework that examines LLM capabilities from multiple complementary angles. Specifically, the design includes: (1) **two semantic formalisms** (SOS and K-semantics) to test robustness across different formalization styles, (2) **three diverse data sources** (Human-Written, LLM-Translated, and Fuzzer-Generated) that span natural programmer style to adversarially challenging edge cases, (3) **nonstandard semantics** (KeywordSwap and KeywordObf) to distinguish true semantic understanding from memorization, and (4) **three tasks of increasing granularity** (PredState, PredRule, PredTrace) that probe different aspects of interpreter functionality. The fuzzer-generated dataset with controlled complexity metrics is particularly innovative, enabling systematic analysis of how specific code properties (control-flow, data-flow, size) affect model performance. This multi-dimensional design provides unprecedented depth and rigor in evaluating semantic reasoning capabilities.

**Weaknesses:**

## Significance

**1. Narrow Framing of Practical Impact:** The paper motivates the work by suggesting that LLMs could serve as interpreters for "rapid prototyping of new programming languages." However, this framing may be overly narrow and does not fully justify the significance of the benchmark. LLMs could potentially contribute to programming language design and development in many other ways that may be more immediately practical and impactful, such as: (1) generating or refining formal semantic specifications from natural language descriptions, (2) automatically generating test cases or oracles based on semantic rules, (3) detecting inconsistencies or ambiguities in language specifications, (4) suggesting optimizations for compiler implementations, (5) providing natural language explanations of semantic rules to aid language designers and users, or (6) assisting in the design of domain-specific languages by reasoning about semantic trade-offs. Given that the results show current LLMs are far from being reliable interpreters (especially on fine-grained tasks), it is unclear why the interpreter capability specifically is positioned as the primary application, rather than these potentially more achievable and valuable use cases.

**3. Questionable Requirement for Rigorous Step-by-Step Interpretation:** The paper evaluates whether LLMs can act as rigorous interpreters by executing programs step-by-step according to formal semantic rules (particularly in the PredRule and PredTrace tasks). However, it is worth questioning whether this level of rigor is necessary or even desirable for practical applications. Human programmers rarely, if ever, mentally simulate programs by explicitly applying formal semantic rules in sequence—instead, they develop high-level intuitions, use informal reasoning, and focus on key program behaviors rather than exhaustive trace execution. If humans can be effective programmers and language designers without this capability, why should we expect or require LLMs to possess it? The paper does not adequately justify why rigorous step-by-step interpretation is the right evaluation target, as opposed to more flexible forms of semantic understanding that might be sufficient for practical tasks.


**3. Unclear Connection to Downstream Coding Tasks:** The paper demonstrates that current LLMs struggle to be precise program executors, particularly on fine-grained tasks like trace prediction. However, it does not establish what implications this has for other coding tasks where LLMs have shown strong performance, such as code generation, debugging, program repair, or code understanding. Failure to be a 100% precise interpreter does not necessarily mean that LLMs are unhelpful for these other tasks—indeed, LLMs are already widely used in practice despite their imperfect semantic reasoning. The paper would benefit from exploring whether interpreter capability correlates with or predicts performance on these downstream tasks. Without this connection, it is difficult to assess the broader significance of the findings beyond the narrow interpreter use case.

> **Suggestion:** It would be valuable to include an empirical study examining the relationship between performance on PLSemanticsBench and performance on established coding benchmarks (e.g., HumanEval, MBPP, or debugging tasks). For instance, do models that perform better on PredTrace also generate more correct code? Do they produce fewer semantic bugs? Such analysis would help establish whether interpreter capability is a fundamental building block for other coding abilities, or whether it is a relatively independent skill with limited practical implications.



## Effectiveness

**1. Limited Coverage of State-of-the-Art Models:** The paper evaluates 11 models, but only one truly state-of-the-art model (GEMINI-2.5-PRO) is included in the evaluation. Notably, other leading models such as GPT-5 series and Claude-4/4.5 series are absent from the study. Given that Gemini-2.5-PRO achieves near-saturated performance on the PredState and PredRule tasks (e.g., 100% on Human-Written PredState under standard semantics as shown in Table 5), it is unclear whether the benchmark has sufficient headroom to differentiate among the most capable models. This raises concerns about the benchmark's ability to track progress as models continue to improve, and whether the observed limitations are fundamental to current LLM architectures or simply a matter of scale and training.

> **Suggestion:** It would be beneficial to evaluate additional state-of-the-art models to better understand the performance ceiling on these tasks. Furthermore, the authors could discuss strategies for maintaining the benchmark's discriminative power over time, such as introducing more challenging program variants or additional task variations that remain difficult even for the best current models.


**3. Insufficient Error Analysis for PredTrace Task:** While the PredTrace task represents the most fine-grained and arguably most important evaluation of interpreter capabilities, the paper provides limited analysis of the types of errors models make on this task. Table 7 shows that all models perform poorly (most near zero accuracy), but there is no breakdown of error types, error distributions, or analysis of where in the execution trace models begin to fail. For instance, do models fail immediately at the first step, or do they correctly predict several steps before diverging? Are certain types of semantic rules more prone to errors than others? Without this deeper analysis, it is difficult to derive actionable insights for improving model performance on this critical task.

> **Suggestion:** It would be highly valuable to include a detailed error analysis for the PredTrace task. Specifically, the authors could provide: (1) a taxonomy of error types (e.g., incorrect rule selection, incorrect state computation, premature termination), (2) the distribution of these error types across different models and program complexities, and (3) a performance curve showing accuracy as a function of execution step number. This would reveal whether models struggle uniformly throughout execution or if performance degrades after a certain number of steps, providing crucial insights into the nature of the limitation.


**3. Artificial Nature of Nonstandard Semantics:** The use of nonstandard semantics (KeywordSwap and KeywordObf) is a clever way to test for true semantic understanding versus memorization. However, these mutations are highly artificial and do not represent the kinds of semantic variations that might occur in the real world. It is unclear whether performance on these tasks is a good predictor of how a model would perform on more realistic semantic challenges, such as dealing with different language versions or domain-specific language extensions.

> **Suggestion:** It might be worthwhile to consider more realistic semantic variations. For example, the authors could introduce subtle changes to the semantics of a language that reflect historical changes in a language's specification (e.g., changes in how variable scoping works in different versions of Python). This would provide a more realistic test of an LLM's ability to adapt to different semantic contexts.

**Questions:**

## Clarification Questions

1.  **Selection of Statements for PredRule Task:** The paper mentions that for the `PredRule` task, a subset of statements is selected from each program for evaluation. Could the authors provide more detail on the selection strategy? For instance, was the selection purely random, or was it stratified to ensure a balanced representation of different types of semantic rules and statement complexities? Understanding this would help in assessing the comprehensiveness of the `PredRule` evaluation.

2.  **High Performance on Complex Programs:** In Table 5, it is noted that some reasoning models achieve very high accuracy on the `PredState` task, even for programs with high complexity (e.g., nested loops of depth five or more). This is a surprising and impressive result. Could the authors elaborate on the characteristics of these complex programs that the models are able to interpret successfully? Specifically, for the reasoning models, are they **using high-level informal reasoning like humans or step-by-step execution simulation**?


3.  **Negative Impact of Formal Semantics:** One of the most intriguing findings is that providing formal semantics can sometimes degrade performance, especially on more complex programs. The paper touches on this briefly, but it would be beneficial to hear the authors' hypotheses on this phenomenon. Is it possible that the formal notation introduces a level of syntactic complexity that distracts the models? Or could it be that the models' pre-trained knowledge of standard programming constructs conflicts with the provided formal rules, leading to confusion?


### Discussion Questions

1.  **Broader Role of LLMs in Language Development:** This work focuses on the role of LLMs as interpreters. Looking beyond this, what are the authors' thoughts on the potential for LLMs to contribute to other aspects of programming language design and implementation? For example, could an LLM be used to automatically generate a formal semantics from a natural language description of a new language, or perhaps even to suggest optimizations for an existing interpreter based on an analysis of its semantic rules?

2.  **The Future of Nonstandard Semantics:** The use of nonstandard semantics is a key innovation of this benchmark. However, as LLMs continue to evolve and are trained on increasingly diverse datasets (which may eventually include formal methods literature and even this paper), there is a risk that they will learn to recognize these
nonstandard patterns as well. What are the authors’ views on the long-term viability of this evaluation strategy? Will it be necessary to continuously develop new and more sophisticated forms of semantic mutation to stay ahead of the models’ learning capabilities?

3.  **Synergy with Other Semantic Reasoning Tasks:** This paper draws a clear distinction between its interpreter-based evaluation and other semantic reasoning tasks like equivalence checking (EquiBench). However, it seems plausible that these different reasoning skills are not entirely independent. For example, an LLM that is good at predicting execution traces might also be better at determining if two programs are equivalent. Do the authors have any thoughts on the potential synergies between these different forms of semantic reasoning? Could training a model on a combination of these tasks lead to a more holistic and robust understanding of program semantics?

4.  **Generalization to Real-World Languages:** The paper acknowledges that "strong performance on PLSEMANTICSBENCH does not necessarily generalize to more sophisticated programming languages." This is a critical point for the practical applicability of the findings. Have the authors conducted any preliminary experiments, even informally, to assess how well the best-performing models on PLSemanticsBench handle real-world programming languages? For instance, if GEMINI-2.5-PRO achieves near-perfect performance on IMP, how does it fare when asked to interpret a simple Python or JavaScript program with provided semantics? Understanding the magnitude of the generalization gap would help the community better contextualize the benchmark results and set realistic expectations for practical applications.

---

### Meta-Review · Area_Chair_ULuS · 2026-01-10

**Summary:**

The paper introduces PLSemanticsBench, a benchmark for testing whether LLMs can execute programs by applying formal operational semantics rather than relying on memorized programming patterns. It uses a small imperative language (IMP) with SOS and K-semantics, evaluates three tasks (final state, rule sequence, full trace), and adds nonstandard semantics via systematic mutations to probe genuine rule-following. Across models, performance is strong on coarse tasks under standard semantics, but drops sharply under mutated semantics and on trace-level prediction.

**Reviewer Concerns:**

The main issues raised are about significance and interpretation: reviewers question whether positioning LLM as interpreter is the right primary application and ask for clearer connections to downstream coding tasks and PL tooling. They also request stronger justification for demanding exact stepwise rule/trace prediction, plus substantially deeper PredTrace error analysis and controls to rule out confounds such as prompt length or evaluation brittleness. Additional concerns include limited external validity due to the toy language, the artificiality of the semantic mutations, and whether the benchmark will retain headroom and model coverage as frontier models improve.

**Reviewer Scores:**

N/A no rebuttal

---

### Decision · Program_Chairs · 2026-01-26

Reject